# Machine learning prediction of the degree of food processing

Giulia Menichetti [1,2], Babak Ravandi [2], Dariush Mozaffarian [3,4] &
Albert-László Barabási [2,5,6] ✉

Despite the accumulating evidence that increased consumption of ultra-processed food has adverse health implications, it remains difficult to decide what constitutes processed food. Indeed, the current processing-based classification of food has limited coverage and does not differentiate between degrees of processing, hindering consumer choices and slowing research on the health implications of processed food. Here we introduce a machine learning algorithm that accurately predicts the degree of processing for any food, indicating that over 73% of the US food supply is ultra-processed. We show that the increased reliance of an individual's diet on ultra-processed food correlates with higher risk of metabolic syndrome, diabetes, angina, elevated blood pressure and biological age, and reduces the bio-availability of vitamins. Finally, we find that replacing foods with less processed alternatives can significantly reduce the health implications of ultra-processed food, suggesting that access to information on the degree of processing, currently unavailable to consumers, could improve population health.

Unhealthy diet is a major risk factor for multiple non-communicable diseases, from obesity and type 2 diabetes to coronary heart disease (CHD) and cancer, together accounting for 70% of mortality and 58% of morbidity worldwide[1,2]. Traditionally, consumers rely on food category-based dietary recommendations like the Food Pyramid (1992[3]) or MyPlate (2011[4]), which define the mix of fruits, vegetables, grains, dairy, and protein-based foods that constitute a healthy diet. In recent years, however, an increasing number of research studies and dietary guidelines have identified the important role and separate health effects of food processing[5–10]. Observational studies, meta-analysis, and controlled metabolic studies suggest that dietary patterns relying on unprocessed foods are more protective than the processing-heavy Western diet against disease risk[11,12]. The role of processed food has reached food policy and is now embodied in several expertise-based food classification systems used in cohort studies such as EPIC[13,14], and led to the expansion of food description and ontology systems such as LanguaL[15], FoodEx2[16], and FoodOn[17]. This

literature indicates a shift from food security, which focuses on access to affordable food, to nutrition security, which emphasizes the need for wholesome and healthful foods[18]. However, known limitations of the current food processing classification systems prompted scientists to advocate for a more objective definition of the degree of processing based on the underlying biological mechanisms rather than on qualitative definitions from different research groups, which challenge the reproducibility of scientific results[13,19]. These observations also align with the growing demand for high-quality and internationally-comparable statistics powered by AI and led by the Food and Agriculture Organization (FAO)[20], as a way to implement and promote objective metrics, reproducibility, and informed decision-making, advancing the convergence towards the United Nations Sustainable Development Goals (SDGs)[21].

NOVA[22–24] is a classification system widely used in epidemiological studies, assessing the extent and purpose of food processing. It categorizes individual foods into four broad categories: *unprocessed or*

[1]Channing Division of Network Medicine, Department of Medicine, Brigham and Women's Hospital, Harvard Medical School, Boston, MA, USA. [2]Network Science Institute and Department of Physics, Northeastern University, Boston, MA, USA. [3]Tufts Friedman School of Nutrition Science and Policy, Boston, MA, USA. [4]Tufts School of Medicine and Medical Center, Boston, MA, USA. [5]Department of Network and Data Science, Central European University, Budapest, Hungary. [6]Department of Medicine, Brigham and Women's Hospital, Harvard Medical School, Boston, MA, USA. ✉e-mail: a.barabasi@northeastern.edu

*minimally processed* (NOVA 1), like fresh, dry, or frozen fruits or vegetables, grains, legumes, meat, fish, and milk; *processed culinary ingredients* (NOVA 2), like table sugars, oils, fats, and salt; *processed foods* (NOVA 3), like canned food, simple bread, and cheese; and *ultra-*

*processed products* (NOVA 4), industrial formulations typically of five or more ingredients including substances not commonly used in culinary preparations, such as additives whose purpose is to imitate sensory qualities of fresh food. Examples of ultra-processed foods

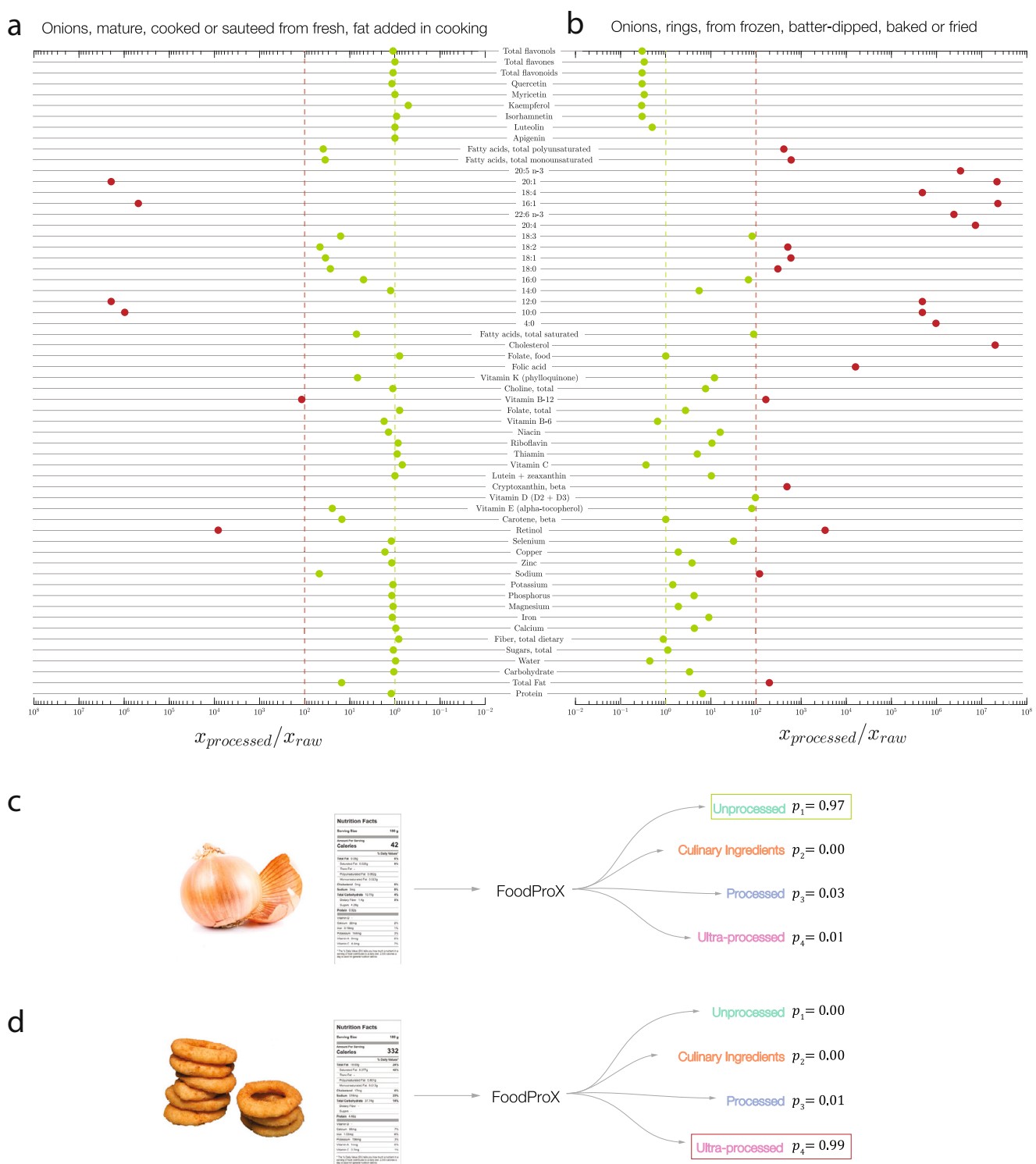

**Fig. 1 | Food processing and nutrient changes (FoodProX). a, b** Ratio of nutrient concentrations for 100 g of Sauteed Onion and Onion Rings compared to Raw Onion, indicating how processing alters the concentration of multiple nutrients. All nutrients in excess of at least two orders of magnitude compared to the concentrations found in the raw ingredient are shown in red. **c, d** We trained Food-ProX, a random forest classifier over the nutrient concentrations within 100 g of each food, tasking it to predict its processing level according to NOVA. FoodProX

represents each food by a vector of probabilities $\{p_i\}$, capturing the likelihood of being classified as unprocessed (NOVA 1), processed culinary ingredients (NOVA 2), processed (NOVA 3), and ultra-processed (NOVA 4). The highest probability determines the final classification label, highlighted in a box on the right. The results shown are for an input list of 99 nutrients. Source data are provided in Source Data Figure 1a–d.xlsx.

include packaged breads, cookies, sweetened breakfast cereals, margarines, sauces and spreads, carbonated drinks, hot dogs, hamburgers, and pizzas.

Epidemiological studies have documented significant associations between greater consumption of NOVA 4 and disease onset[25–31], including links to obesity[32], CHD[30,33], diabetes mellitus[34,35], cancer[29,36], and depression[28]. Finally, a randomized controlled metabolic trial in 20 adults has confirmed the short-term adverse effects on caloric intake and weight gain of ultra-processed foods[31].

Despite its success, NOVA is qualitative in nature, aiming to address a laborious and incomplete classification task, hence resulting in inconsistencies and ambiguities across the literature[19], and limiting research into the impact of processed food[19,37,38]. First, NOVA relies on expertise-based manual evaluation of each food, and assigns a unique class to only 35% of the foods cataloged by the US Department of Agriculture (USDA), decomposing the remaining part of the database in ingredients to be further analyzed (Section S1). Second, the classification of composite recipes, products, and mixed meals including different types of processed items - a large proportion of the food supply—is not straightforward. Whilst, an accurate ingredient list of all complex foods would allow for an ingredient-based classification, a full and quantified ingredient list is seldom, if ever, available to the consumer, and when available, it shows considerable variability from database to database. Interestingly, even in presence of detailed ingredient information, the consistency across nutrition specialists in assigning NOVA classes was found to be low[38]. Given the lack of well-regulated data on food labels indicating food processes and their purpose, current approaches have classified as ultra-processed all foods with at least one ingredient rarely used in kitchens or with at least one cosmetic additive[19,39], hence unable to discriminate such foods in relation to health outcomes[40,41]. Third, all the observed risk for the NOVA classification is in the NOVA 4 class, representing a large and heterogeneous category of ultra-processed food that limits our ability to investigate the health implications of different gradations of processing. Indeed, according to NOVA, many nations derive up to 60% or more of their average caloric intake from ultra-processed foods[41–43]. While NOVA allows for a more refined analysis, epidemiological and clinical studies have only focused on NOVA 4 as a whole. These perceived homogeneity of NOVA 4 foods limits both scientific research and practical consumer guidance on the health effects of differing degrees of processing. It also reduces the industry's incentives to reformulate foods towards less processed offerings, shifting investments from the ultra-processed NOVA 4 foods to the less processed NOVA 1 and NOVA 3 categories.

Here, we introduce FoodProX, a machine-learning classifier that takes as input nutritional measures, and is trained to predict the degree of processing of any food in a reproducible, portable, and scalable fashion. We rely on nutrients as input because: (1) The list of nutrients in a food is consistently regulated and reported worldwide; (2) Their quantities in unprocessed food are constrained by physiological ranges determined by biochemistry[44]; (3) Food processing systematically and reproducibly alters nutrient concentrations through combinatorial changes detectable by machine learning (Fig. 1a, b). FoodProX allows us to define a continuous index (FPro) that captures the degree of processing of any food, and helps us quantify the overall diet quality of individuals, ultimately unveiling the statistical correlations between the degree of processing characterizing individual diets and multiple disease phenotypes.

## Results
The manual procedure behind NOVA, relying on the ingredients of food, has allowed straightforward labeling to 2484 foods reported in the National Health and Nutrition Examination Survey (NHANES) 2009–2010, representing 34.25% of items consumed by a representative sample of the US population[26] (Section S1). The remaining foods are either not classified, or require further decomposition into their differing food ingredients, often not reported by the manufacturers. In contrast, the basic nutrient profile of foods and beverages is always disclosed, and mandated by law in most countries. For example, the USDA Standard Reference database (USDA SR Legacy), catalogs the nutrient profile of 7793 foods with resolutions ranging from 8 to 138 nutrients (Figure S1)[45], and USDA FNDDS reports between 65 to 102 nutrients for all foods consumed by NHANES participants[46,47].

Our work relies on the hypothesis that the nutrient profiles of unprocessed or minimally processed foods are generally constrained within common physiologic ranges[44]. The nutrient profile can be widely altered by the physical, biological, and chemical processes involved in food preparation and conservation, thus correlating with the degree of processing undertaken. Indeed, among the nutrients reported in raw onion, 3/4 change their concentration in excess of 10% in the fried and battered version of the product, and more than half by 10-fold (Fig. 1b). We lack however, a single nutrient "biomarker" that accurately tracks the degree of processing; instead we observe changes in the concentration of multiple nutrients, whose combinations jointly correlate with processing. This complexity advocates for the use of machine learning, designed to capture the combinatorial explosion of nutrient alterations.

## FoodProX algorithm
To train FoodProX, a multi-class random forest classifier, we leveraged the nutrient concentrations provided in FNDDS 2009–2010 for the foods classified in ref. 26 (Fig. 1d). FoodProX takes as input the list of nutrients in any food and offers as output a vector of four probabilities $\{p_i\}$, representing the likelihood that the respective food is classified as unprocessed ($p_1$, NOVA 1), processed culinary ingredients ($p_2$, NOVA 2), processed ($p_3$, NOVA 3), and ultra-processed ($p_4$, NOVA 4). The highest of the four probabilities determines the final classification label for each food (Fig. 1c, d).

As 90% of foods in the USDA Branded Food Products Database report less than 17 nutrients, we also tested the algorithm's predictive power with the 12 gram-based nutrients mandated by the FDA[48] (Section S1). To evaluate the performance of FoodProX we measure the area under the receiver operating characteristic (AUC), defined as the probability that a random sample from the class of interest will have a higher score than a random sample from any other class. We identified consistently high AUC values across all the considered nutrient subsets: $0.9804 \pm 0.0012$ for NOVA 1, $0.9632 \pm 0.0024$ for NOVA 2, $0.9696 \pm 0.0018$ for NOVA 3, and $0.9789 \pm 0.0015$ for NOVA 4, significantly far from a random performance with AUC = 0.5, describing a model with no discriminating power (see Section S2.1, Figure S5, and Table S6 for a detailed analysis of the cross-validated performances, including precision and recall). These results confirm that changes in nutrient content have remarkable predictive power in capturing the extent of food processing. Furthermore, we find that no single nutrient drives the predictions, but the predictive signal is rooted in combinations of changes spanning multiple nutrients (see Section S2.2 for a detailed permutation feature importance and Shapley value analysis[49]).

We visualize the decision space of the classifier by performing a principal component analysis over the probabilities $\{p_i\}$, observing that the list of foods manually classified by NOVA is limited to the three corners of the phase space, to which the classifier assigns dominating probabilities (Fig. 2a). We used FoodProX to classify all foods and beverages that lacked prior manual NOVA classification in FNDDS (65.75% of the total). We found that 7.39% of FNDDS consists of NOVA 1; 0.90%, NOVA 2; 18.36%, NOVA 3; and 73.35%, NOVA 4 foods (Fig. 2b). Yet, many previously unclassified foods are often inside the phase space, indicating that they lack a dominating

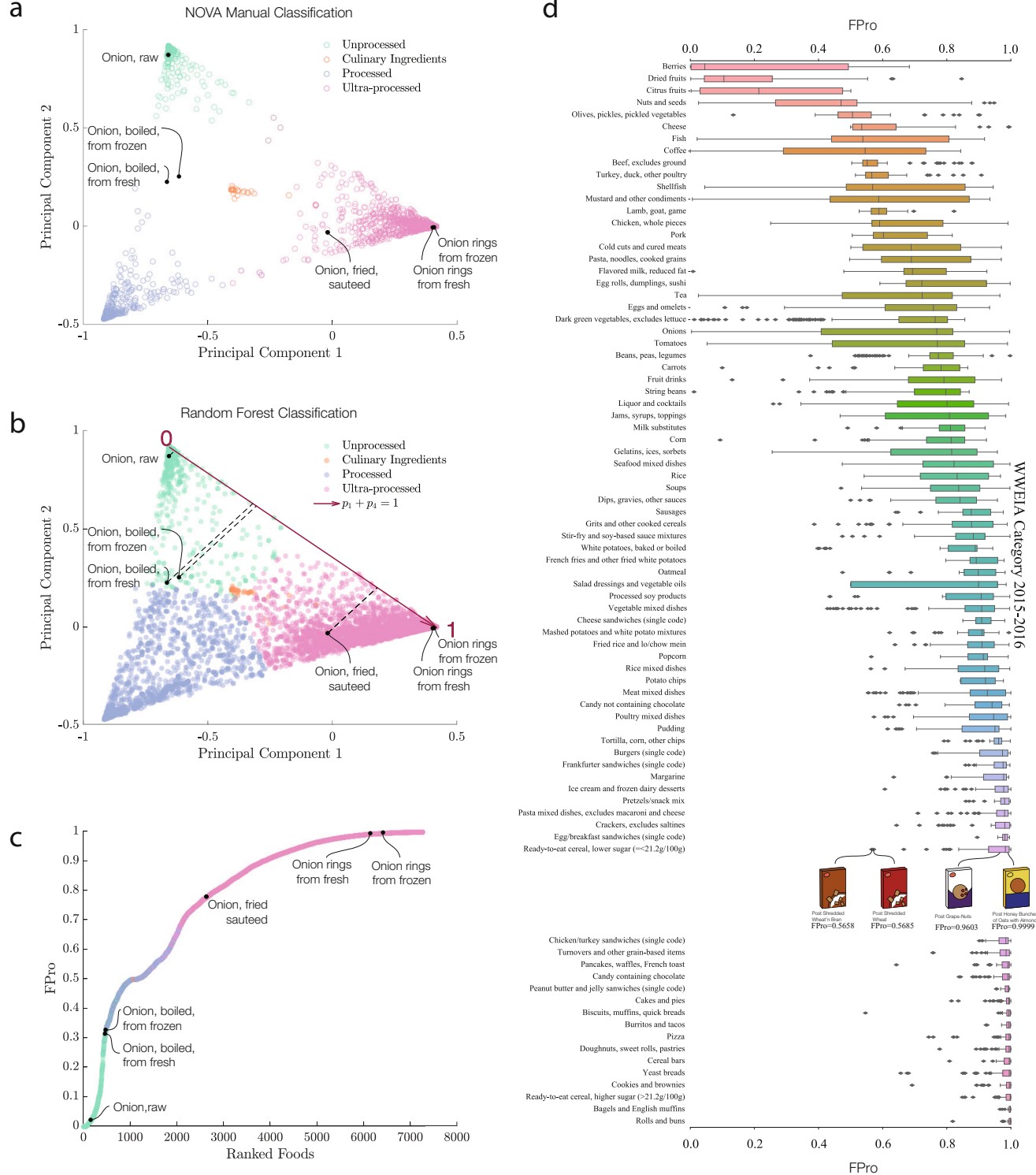

probability, hence the assignment of a single NOVA class is somewhat arbitrary (Fig. 2b). The detection of this ambiguity is a strength of FoodProX, reflecting the observation that a four-class classification encodes a great extent of nutrient variability associated with different food processing methods and intensities[50]. For example, while the classifier confidently assigns "Raw Onion" to NOVA 1 ($p_1 = 0.9651$), and "Onion rings prepared from frozen" ($p_4 = 0.9921$) to NOVA 4, it accurately offers an intermediate confidence for "Onion, Sauteed," placing it with probability $p_4 = 0.6521$ in NOVA 4, and with probability $p_3 = 0.2488$ in NOVA 3.

## Food processing score (*FPro*)

The observation that enforcing discrete classes causes inherent challenges in food classification prompted us to introduce the food processing score (*FPro*), a continuous variable with *FPro* = 0 for raw ingredients, and *FPro* → 1 for ultra-processed foods. We define the *FPro* of a food $k$ as

$$FPro_k = \frac{1 - p_1^k + p_4^k}{2}, \tag{1}$$

**Fig. 2 | NOVA classification and processing score. a** Visualization of the decision space of FoodProX via principal component analysis of the probabilities $\{p_i\}$. The manual 4-level NOVA classification assigns unique labels to only 34.25% of the foods listed in FNDDS 2009–2010 (empty circles). The classification of the remaining foods remains unknown, or must be further decomposed into ingredients. The list of foods manually classified by NOVA is largely limited to the three corners of the phase space, foods to which the classifier assigns dominating probabilities. **b** FoodProX assigned NOVA labels to all foods in FNDDS 2009–2010. The symbols at the boundary regions indicates that for these foods the algorithm's confidence in the classification is not high, hence a 4-class classification does not capture the degree of processing characterizing that food. For each food $k$, the processing score $FPro_k$ represents the orthogonal projection (black dashed lines) of $\vec{p}^{\,k} = (p_1^k, p_2^k, p_3^k, p_4^k)$ onto the line $p_1 + p_4 = 1$ (highlighted in dark red). **c** We ranked all foods in FNDDS 2009/2010 according to $FPro$. The measure sorts onion products in increasing order of processing, from "Onion, Raw", to "Onion rings, from frozen". **d** Distribution of $FPro$ for a selection of the 155 Food Categories in What We Eat in America (WWEIA) 2015–2016 with at least 20 items (Section S2). WWEIA categories

group together foods and beverages with similar usage and nutrient content in the US food supply[52]. Sample sizes vary from a minimum of 21 data points for "Citrus fruits" to a maximum of 340 data points for "Fish". For each box in the box plots, the minimum indicates the lower quartile, the central line represents the median, and the maximum corresponds to the upper quartile. The upper and lower whiskers represent data outside of the inter-quartile range. All categories are ranked in increasing order of median $FPro$, indicating that within each food group, we have remarkable variability in $FPro$, confirming the presence of different degrees of processing. We illustrate this through four ready-to-eat cereals, all manually classified as NOVA 4, yet with rather different $FPro$. While the differences in the nutrient content of Post Shredded Wheat 'n Bran ($FPro = 0.5658$) and Post Shredded Wheat ($FPro = 0.5685$) are minimal, with lower fiber content for the latter, the fortification with vitamins, minerals, and the addition of sugar, significantly increases the processing of Post Grape-Nuts ($FPro = 0.9603$), and the further addition of fats results in an even higher processing score for Post Honey Bunches of Oats with Almonds ($FPro = 0.9999$), showing how $FPro$ ranks the progressive changes in nutrient content. Source data are provided in Source Data Figure 2a–d.xlsx.

capturing the progressive changes in nutrient content induced by processing (see Box 1). In other words, the $FPro$ score measures the trade-off between the confidence that the FoodProX algorithm has in classifying food item $k$ as NOVA 1 ($p_1^k$) or NOVA 4 ($p_4^k$), which are the two extreme classes clearly ranked according to an increasing extent of food processing[19,51]. For example, $FPro$ incrementally increases from raw onion ($FPro = 0.0203$) to boiled onion ($FPro = 0.3126$), fried onion ($FPro = 0.7779$), and onion rings from frozen ingredients ($FPro = 0.9955$, Fig. 2c). $FPro$ allows us to unveil the degree of processing characterizing different food preparation techniques, assigning lower values to foods made from fresh ingredients than those made from frozen ingredients (Fig. 2c). $FPro$ also permits classification of complex recipes and mixed dishes, identifying the overall average processing of the item or meal.

As noted above, nearly 20% of the US diet comprises foods in NOVA 3, and over 70%, in NOVA 4. The observed variability of $FPro$ for foods that belong in the same NOVA class prompted us to analyze the variations in degree of processing within subgroups of foods and beverages, as captured by the What We Eat in America (WWEIA) categorization[52]. We find that different food products within the same NOVA classification and WWEIA categories show remarkable variability in $FPro$, confirming the presence of different processing fingerprints within each category (Fig. 2d). For example, if we select four products from the same brand (Post) of breakfast cereals that are all manually classified as NOVA 4 (i.e., considered identically processed), we observe that $FPro$ differentiate them and captures progressive alterations in fiber content, fortification with vitamins and minerals, and addition of sugar and fats (Fig. 2d, insert, and Section S2.5).

**Stability and robustness of $FPro$**

Real-world foods vary in nutritional content, affected by recipe variations, production methods, soil quality, storage time, and changing government regulations[53]. This degree of variability, coupled with measurement and reporting uncertainties, raises a fundamental question: is $FPro$ robust against the expected variability and uncertainty in nutrient content? To address this, we first explored how nutritional values for the same food change through different FNDDS cycles, focusing on the 5,632 foods whose nutrient profile is reported both in FNDDS 2009–2010 and 2015–2016 (Section S7). For example, in the less processed category, for "Milk, calcium fortified, cow's, fluid, whole" we find that 14 nutrients have changed between 2009 and 2015, including a 3.65-fold decrease in Calcium content. In a similar fashion, the highly ultra-processed "Cookie, vanilla with caramel, coconut, and chocolate coating" shows variation in 46 nutrients, with 6 fold decrease in monounsaturated fatty acid 20:1. Despite such significant changes in the content of some nutrients, $FPro$ shows remarkable stability: milk's $FPro$ goes from 0.0010 to 0.0011, consistently

classifying it as unprocessed, and for the cookie $FPro$ goes from 0.9943 to 0.9965, staying firmly in the ultra-processed category. Overall, for foods whose nutrients maximally change between 10% and 50% of their original value, we observe an absolute shift in $FPro$ of 0.001556 (quartiles $Q_1(25\%) = 0.000222$, $Q_3(75\%) = 0.004764$). Allowing up to 1000% of nutrient variability does not significantly alter our findings, since the expected change of $FPro$ per food reaches 0.003312 (quartiles $Q_1(25\%) = 0.000722$, $Q_3(75\%) = 0.011310$). The observed $FPro$ stability is rooted in the fact that $FPro$'s value is driven by the nutrient panel as a whole, and not by the concentration of any single nutrient (see Section S7, for further details on data sampling and variability).

Currently, the chemical information available to train $FPro$ does not track the concentration of additives as they are rarely available for most foods. Yet, additives like tertiary butylhydroquinone, acetylated monoglycerides, polysorbates, sodium stearoyl lactylate, and sodium aluminum phosphate, represent obvious signatures of food processing[54], raising the question on how much improvement in predictive power we could obtain if the information on additives would be available. We, therefore, relied on Open Food Facts, that compiles an extensive list of food additives, including artificial colors, artificial flavors, and emulsifiers[55], to test the $FPro$'s ability to absorb information on additives. From the Open Food Facts website we collected 233,831 nutritional records, annotated with NOVA labels according to a heuristic described in ref. 56. We then trained and validated two models: (1) FoodProX leveraging only nutrition facts as introduced before, (2) a modified FoodProX, using nutrition facts and the available information on the number of additives (Section S6). While model (2) displays slightly better performance, with AUC $0.9926 \pm 0.0003$ for NOVA 1, $0.9878 \pm 0.0047$ for NOVA 2, $0.9653 \pm 0.0010$ for NOVA 3, $0.9782 \pm 0.0007$ for NOVA 4, we find that the performance of model (1) is largely indistinguishable, reaching AUC $0.9880 \pm 0.0006$ for NOVA 1, $0.9860 \pm 0.0045$ for NOVA 2, $0.9320 \pm 0.0015$ for NOVA 3, $0.9508 \pm 0.0009$ for NOVA 4. In other words, while information on additives can improve FoodProX's performance, changes in the nutrient profile already carry the bulk of the predictive power. This aspect is further confirmed by the performance of a classifier purely based on the number of additives: in this scenario the number of false positives for NOVA 1, 2, and 3 remarkably increases, affecting the precision of the model, and showing predictive power only for NOVA 4 (see Section S6 for a detailed comparison of the models).

Finally, we investigate how $FPro$, a measure of the nutritional quality of food, changes depending on where the food was prepared, exploring if it can distinguish between home-cooked food, food prepared in stores, canteens, restaurants, fast foods, and products available in vending machines. Leveraging data from NHANES, we ranked the 10 most popular food sources in increasing value of $FPro$. The

# BOX 1
# Overview of FoodProX and FPro

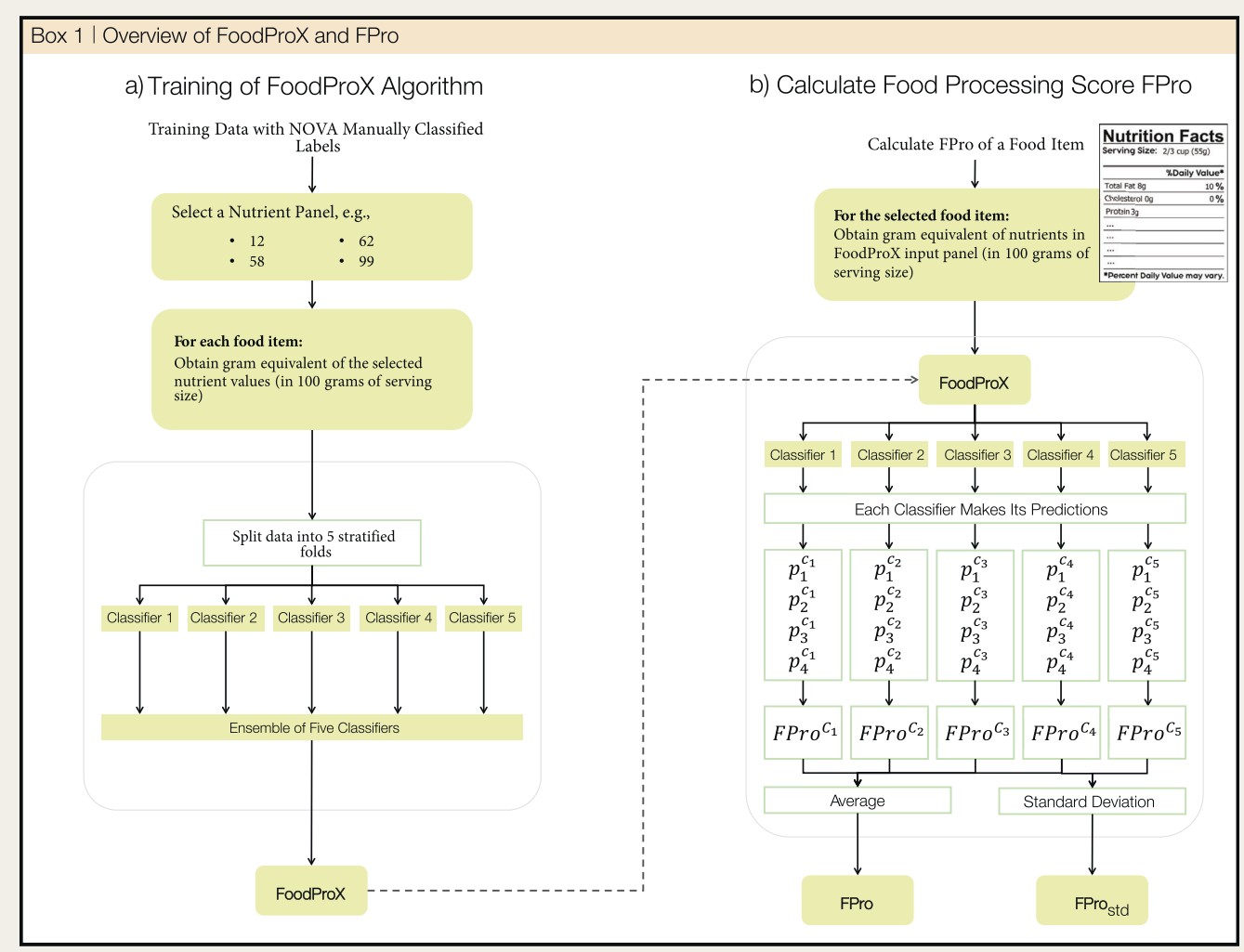

**Schematic overview of the link between FoodProX classifier and *FPro* score (a)** To construct FoodProX, a labeled training dataset with NOVA classes and input nutrient information per 100 grams is first selected. FoodProX is then created as an ensemble voting system that includes five random forest classifiers, each trained on 4/5 of the stratified dataset. Food classification predictions are made based on the average probabilities per class across the five classifiers. **(b)** To calculate *FPro* for a specific food item, an input list of nutrients compatible with the trained FoodProX is required. For each classifier in the ensemble, *FPro* is calculated using Eq. (1), which enables us to estimate the average and standard deviation across the models. For further details see the Methods Section.

lowest *FPro* is observed for "Grown or caught by you and someone you know" (median *FPro* = 0.4423) and "Residential dining facility" (median *FPro* = 0.6238), confirming the less processed nature of home-prepared foods. We obtain a much higher *FPro* for "Restaurant fast food/pizza" (median *FPro* = 0.9060) and "Vending machine" (median *FPro* = 0.9800), confirming its reliance on ultra-processed ingredients. This result indicates that *FPro* can differentiate food sources, styles and quality of food preparation (see Section S2.6 for the statistical analysis).

## Individual processing score

The significant contribution of ultra-processed food to American dietary intake, and *FPro*'s ability to demonstrate heterogeneity in the extent of food processing within the broad category of ultra-processed food, prompts us to assess the contribution of processed food to the

diet of each individual, jointly weighted by both the extent of processing and the contribution to caloric intake. This is provided by the individual Food Processing Score (*iFPro*),

$$iFPro^j_{WC} = \sum_k^{D_j} \frac{c^j_k}{C^j} FPro_k, \tag{2}$$

which varies between 0 and 1, where $D_j$ is the number of dishes consumed by individual $j$, $C^j$ is the daily total amount of consumed calories, and $c^j_k$ is the amount of calories contributed by each food item. A gram-based *iFPro_WG*, captures the fraction of grams in a diet supplied by processed food (Eq. S6).

We calculated *iFPro* for 20,047 individuals with dietary records in a representative US national sample from NHANES 1999–2006.

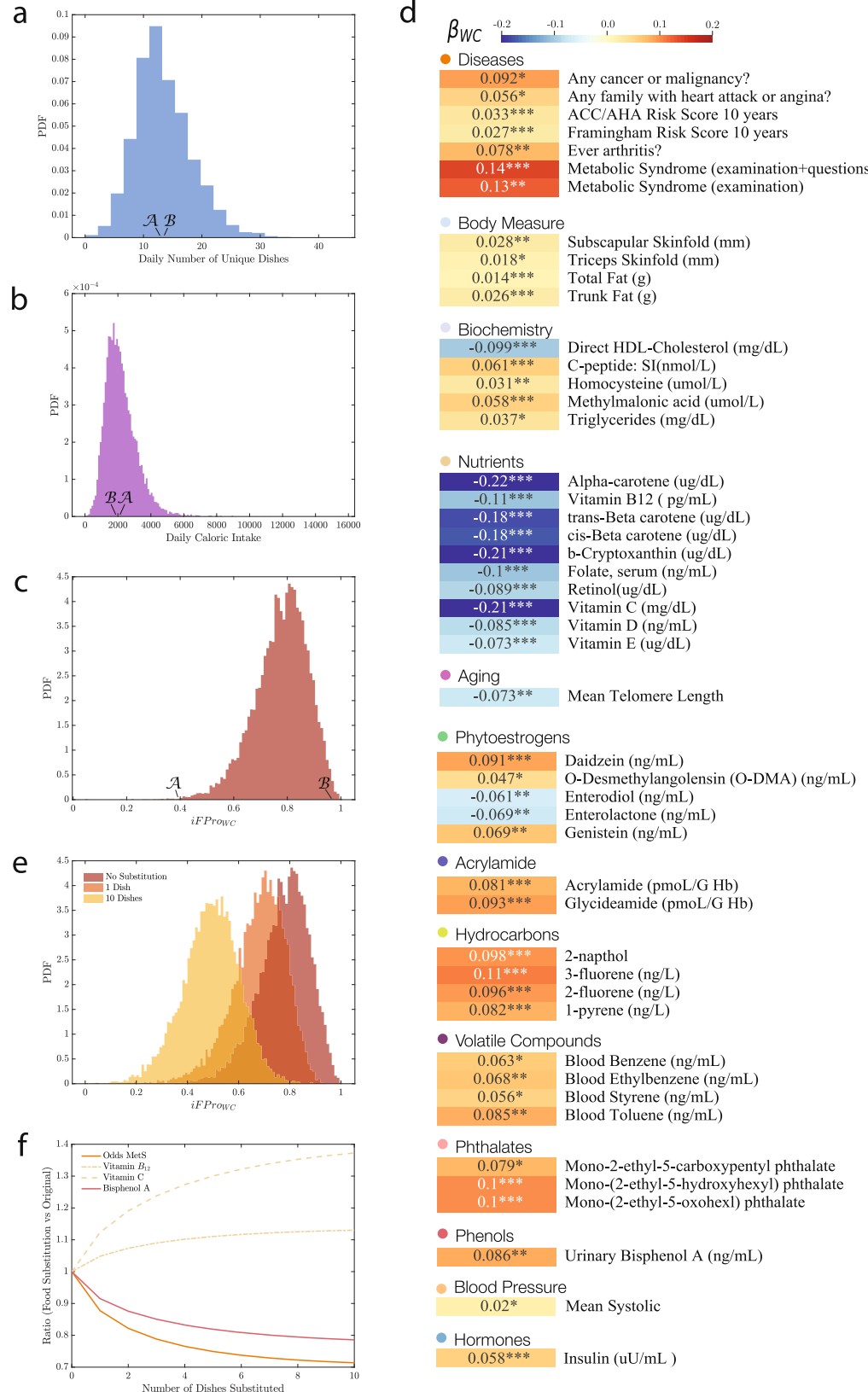

NHANES relies on 24-hour recalls to capture dietary intake, a widely used methodology in epidemiology, with well-explored statistical characteristics[57]. Indeed, studies collecting multiple days of intake data per individual indicate that it is statistically more efficient to increase the number of individuals in the sample than increase the number of days beyond two per individual[58]. As Fig. 3c shows, the median *iFPro_WC* for the American population is 0.7872, confirming a high reliance on intake on ultra-processed food[41–43]. More importantly, *iFPro* allows us to identify differential reliance on processed food. Consider individuals $\mathcal{A}$ and $\mathcal{B}$, men of similar age (47 vs. 48 years old, Fig. 3a–c), with

**Fig. 3 | Health implications and food substitution.** For each of the 20,047 individuals in NHANES (1999–2006), 18+ years old with dietary records[59], we calculated the individual diet processing scores $iPro_{WC}$. **a** The average number of unique dishes reported in the dietary interviews, highlighting two individuals $\mathcal{A}$ and $\mathcal{B}$, with comparable number of dishes, 12.5 and 13 reported, respectively. **b** The distribution of average daily caloric intake, showing that individuals $\mathcal{B}$ and $\mathcal{A}$ have similar caloric intake of 1894 and 2016 kcal, respectively. **c** The distribution of $iPro_{WC}$ for NHANES, indicating that individuals $\mathcal{A}$ and $\mathcal{B}$ display significant differences in $iPro_{WC}$, with $\mathcal{B}$'s diet relying on ultra-processed food ($iPro_{WC} = 0.9572$), and $\mathcal{A}$ reporting simple recipes ($iPro_{WC} = 0.3981$) (Figure S13). **d** We measured the association of various phenotypes with $iPro_{WC}$, correcting for age, gender, ethnicity, socioeconomic status, BMI, and caloric intake (Section S4). We report the standardized $\beta$ coefficient, quantifying the effect on each exposure when the Box-Cox transformed dietary scores increase by one standard deviation over the population.

For continuous exposures the coefficients are fully standardized, while for logistic regression (disease phenotypes) we opted for partially standardized coefficients to help interpretability (Section S4). Each variable is color-coded according to $\beta$, positive associations shown in red, and negative associations in blue. For logistic regressions, $p$ values are associated with two-sided Wald tests, while for multiple linear regressions, $p$ values are determined by two-sided $t$ tests. Here, we show a selection of the 209 variables surviving Benjamini-Hochberg FDR correction with $\alpha = 0.05$ (*** adj $p$ value < 0.001, ** adj $p$ value < 0.01, * adj $p$ value < 0.05) **e** Changes in $iPro_{WC}$ when one (orange) or up to ten (yellow) dishes are substituted with their less processed versions, following the prioritization rule defined in Eq. S8. **f** The impact of substituting different number of dishes on the odds of metabolic syndrome, concentrations of vitamin $B_{12}$, vitamin C, and bisphenol A, showing that a minimal substitution strategy can significantly alter the health implications of ultra-processed food. Source data are provided in Source Data Figure 3a–f.xlsx.

similar number of reported daily unique dishes (12.5 vs. 13 dishes) and comparable caloric intake (2016 vs. 1894 kcal). Yet, the diet of individual $\mathcal{A}$ has $iPro_{WC} = 0.3981$, as it relies on unprocessed ingredients and home cooking. In contrast, $\mathcal{B}$ has $iPro_{WC} = 0.9572$, given his consumption of hot dogs, hamburgers, French fries, pizza, and Kit Kat (see Section S3.5 for a full analysis of the correlation between $iPro_{WC}$ and food group consumption in the US population). As we show next, these differences correlate with different health outcomes.

## Health implications

To quantify the degree to which the consumption of ultra-processed foods, graded across their entire range, correlate with health outcomes, we investigated exposures and phenotypes provided in NHANES 1999–2006[59]. Inspired by ref. 60, we performed an Environment-Wide Association Study (EWAS), an alternative to single-exposure epidemiological studies, to identify the most significant environmental factors associated with different health phenotypes in a non-hypothesis-driven fashion. We measured for each variable the association with the diet processing score $iPro$, adjusting for age, sex, ethnicity, socioeconomic status, BMI, and caloric intake (Figure S17). After False Discovery Rate (FDR) correction for multiple testing, 209 variables survive (Section S4), documenting how the consumption of ultra-processed foods relates to health.

**Metabolic and cardiovascular risk.** We find that individuals with a high food processing score show positive associations with risk of metabolic syndrome (MetS), diabetes (fasting glucose), and also (consistent with poor diets clustering in families) a family history of heart attack or angina, in line with earlier findings (Section S4)[60–62]. The increased risk of cardiovascular disease is further confirmed by the significant positive association of $iPro$ with both Framingham and ACC/AHA Risk Scores (Fig. 3d)[63,64].

Overall, individuals with a higher food processing score exhibit higher blood pressure, trunk fat, and subscapular skinfold, measures of obesity, blood insulin, and triglyceride levels; and lower "good" HDL cholesterol. Further novel findings indicate a higher prevalence of type 2 diabetes (C-peptide), inflammation (C-Reactive Protein), vitamin deficiency (homocysteine, methylmalonic acid), and inflammatory arthritis[65–70]. We also find an inverse association between $iPro$ and telomere length, which can be affected by diet through inflammation and oxidation[71], suggesting a higher biological age for individuals with higher reliance on more ultra-processed foods (Section S4, Figure S21).

**Vitamins and phytoestrogens.** A greater consumption of more extensively processed foods correlates with lower levels of vitamins in our bloodstream, like vitamin $B_{12}$ and vitamin C, despite the fact that ultra-processed breakfast cereals and refined flours are frequently fortified with these vitamins and minerals. In addition, $iPro$ allows us to discriminate plant-based diets relying on legumes, whole grains, fruits, and vegetables, from diets that include ultra-processed plant-

based meat, dairy substitutes, and plant-based drinks[72]. For example, we observe a positive correlation between $iPro_{WC}$ and the urine levels of daidzein, genistein, and their bacterial metabolite o-desmethylangolensin—all bio-markers of soy, abundant in diets that rely on highly processed soy protein foods. On the other hand, enterolactone and enterodiol, gut metabolites of plant lignans (fiber-associated compounds found in many plant families and common foods, including grains, nuts, seeds, vegetables, and drinks such as tea, coffee, or wine) are inversely associated with $iPro_{WC}$[73].

**Chemical exposures.** Diet rich in highly processed food shows an association with increased carcinogenic, diabetic, and obesity-inducing food additives and neoformed contaminants, several of which represent previously unknown relationships. For instance, we find positive associations with acrylamide and polycyclic aromatic hydrocarbons, present in heat-treated processed food products as a result of Maillard reaction, benzenes (abundant in soft drinks), furans (common in canned and jarred foods), PCBs (processed meat products), perfluorooctanoic acids and phthalates (found in the wrappers of some fast foods, microwavable popcorn, and candies), and environmental phenols like the endocrine disruptor bisphenol A (linked to plastics and resins for food packaging)[74–79]. Importantly, all these represent compounds not reported in food composition databases, but recovered in blood and urine (Figure S20).

Taken together, our ability to distinguish the degree of processing of foods in individual diets allowed us to unveil multiple correlations between diets with greater reliance on ultra-processing and health outcomes. This approach leads to results that cannot be captured with the existing NOVA classification (Section S4, Figures S18 and S19). Specifically, if we rely on the manual NOVA 4 and the panel of exposures in NHANES, we recover only 92 significant associations. Among the missing associations are the inverse correlation of a more ultra-processed diet with vitamin D and folate blood levels, and the positive correlation with c-peptide, homocysteine, and blood pressure (Section S4).

## Food substitution

The high reliance of ultra-processed food among the US population (Fig. 3c) prompts us to ask: what kind of interventions could help us reduce the observed health implications? Given the challenges of large behavioral shifts[80,81], here we assume that an individual does not need to overhaul her entire diet, but replace the most processed items she consumes with less processed versions of the same item. To minimize the dietary shifts required, we identify within each individual's diet the item with high caloric contribution and for which there are significantly less processed alternatives (Eq. S8), preserving the broad food class (WWEIA) of the initial choice (Section S5). For example, we replace Kix cereals ($FPro = 0.9998$) with shredded wheat and bran cereals ($FPro = 0.5091$), and spread cheese from a pressurized can ($FPro = 0.9648$) with provolone cheese ($FPro = 0.5001$). We find that

the isocaloric replacement of a single item from the average of 19.28 food items daily consumed by US adults reduces the median $iFPro_{WC}$ by 12.15%, from an overall score of 0.7872 to 0.6915 (Fig. 3E). Based on observed associations with phenotypes, this translates to a decrease in the odds of metabolic syndrome by 12.25%, a lower concentration of urinary bisphenol A by 8.47%; and an increased blood concentration of vitamin $B_{12}$ and vitamin C by 4.83% and 12.31%, respectively (Fig. 3f and Eqs. S9 and S10). Furthermore, the substitution of 10 food items, about half the daily reported items, leads to 37.03% decrease in $iFPro_{WC}$, with associated changes of −21.43% for bisphenol A, +13.02% for blood vitamin $B_{12}$, and +37.26% for blood vitamin C.

Overall, we find that modest substitution strategies made possible by *FPro* can preserve the general nature of an individual's diet but reduce reliance on more highly processed food and successfully moderate the observed health implications. As a first step to implement such strategies in the real world, consumers and policymakers must be empowered with information on the degree of processing characterizing the foods in their food environment available to consume, and their potential alternatives. To successfully and sustainably improve population diets, interventions will need to combine individual nudges and behavior change strategies with interventions that more precisely address the ecological, system-level phenomena that people are exposed to, including addressing major gaps and disparities in life-style and healthy food access (i.e., nutrition security).

## Discussion

In this paper, we introduced *FPro*, a continuous processing score combining in non-linear fashion features of processing techniques learned from the NOVA manual labels, with nutrient concentrations from food composition data. *FPro* is derived from FoodProX, a classifier that shows a remarkable ability to replicate the manual NOVA classification from nutritional information, confirming that NOVA classes result in distinct patterns of nutrient alterations, accurately detected by machine learning.

Importantly, FoodProX allows us to build upon and extend the current NOVA classification in several crucial respects, offering automated and reproducible classification of foods across multiple national and commercial databases, the ability to classify complex recipes and mixed foods and meals, and the ability to quantify the extent of food processing among the large and otherwise homogeneously categorized groups of ultra-processed foods. Given that our algorithm only needs the Nutrition Facts, information already accessible to consumers on packaging and via smartphone apps, web portals, and grocery store and restaurant websites, *FPro* can help monitor the reliance of an individual's diet on less or more processed food. Building on the portability of our model, we were able to extend the *FPro* assessment to over 50,000 products collected from major US grocery store websites, a first step towards a systematic characterization of the food environments[82].

Differently from dietary indexes such as HEI-15[83], designed to measure the alignment of individuals' diets with the 2015–2020 Dietary Guidelines for Americans, *iFPro* and *FPro* help us identify which foods to substitute, to shift individual consumption patterns towards a less processed diet. Our substitution heuristic indicates that minimal changes in diet can significantly reduce disease risk, a strategy hard to implement with the current NOVA classification, which classifies >70% of the food supply as NOVA 4[84].

The consistent predictive power of *FPro* in epidemiological analysis indicates that it offers an accurate global scale of food processing, capturing the chemical-physical alterations of food and its impact on health. However, *FPro* is currently best suited to rank foods within the same food category, hence offering accurate input for substitution strategies, as explored above. In other words, we should first identify a chemically-driven food group (e.g., "Fruit"), and then quantify the

extent of nutrient alterations that leads to different degrees of processing.

Overall, a combination of *FPro* with epidemiological studies and food classification could lead to an automated and practical pipeline capable of systematically improving population diet and individual health. Furthermore, the systematic addition of chemical concentrations for additives and processing byproducts in all foods will enable the construction of an *FPro* that is completely unsupervised and independent from any manual classification. A wider range of chemical classes will also enable a progressively better modeling of the "food matrix effect", capturing the processing and cooking induced transformations in the cellular matrices of plants and muscle tissues.

## Methods

### Training data

The Food and Nutrient Database for Dietary Studies (FNDDS) is a database created by the United States Department of Agriculture (USDA) that provides comprehensive food composition data, such as the amount of vitamin C per 100 g of a selected ingredient, for foods and beverages consumed as part of the National Health and Nutrition Examination Survey (NHANES), which is a biannual cross-sectional survey conducted by the Center for Disease Control and Prevention (CDC) to monitor the health of the American population. Unlike the USDA Standard Reference Legacy (SR) and Foundation Foods (FF), which are designed to disseminate food composition data, FNDDS is specifically tailored to facilitate the analysis of dietary intake. As such, FNDDS contains no missing nutrient values, making it an ideal resource for training machine-learning models[85]. For the years 2007–2010 the USDA created a flavonoids database that expanded the nutritional panel of population surveys from the original 65 nutrients to 102. In our analysis, we focused on all nutrients explicitly measured in grams (g), milligrams (mg), or micrograms (μg), resulting in 99 nutrients. This approach allowed us to focus on a comprehensive set of nutrients while ensuring consistency and comparability across the different nutrients.

We selected FNDDS 2009–2010 as the main data source for training FoodProX because it allowed us to combine the NOVA labels assigned by Steele et al. in[26] with one the most comprehensive nutrient panels available for population studies. Among the 7253 foods included in FNDDS 2009–2010, 2484 food items were originally categorized under a unique NOVA class, while the remaining 4769 foods either lacked classification (730) or required further decomposition (4039) into 2946 ingredients obtained from the SR24 database.

The availability of a large nutritional panel in FNDDS 2009–2010 enabled us to train FoodProX using various subsets of nutrients. The widest panel encompasses 99 nutrients, including flavonoid measurements developed for NHANES 2007–2010[86]. From these 99 nutrients, we selected 62 nutrients that are commonly documented in NHANES 2001–2018, and 58 nutrients that are available across NHANES 1999–2018 cycles, which determined the panel used for our epidemiological analysis utilizing NHANES data from 1999 to 2006. Furthermore, to address the needs of the consumer space and branded products, we trained FoodProX on a specific subset of 12 nutrients that contribute to FDA nutrition facts. Notably, we excluded calories and the total amount of trans fatty acids from this subset, as the latter is not available in the original batch of 99 nutrients.

For further details, see Section S1.

### FoodProx and *FPro*

FoodProX is a machine-learning model that leverages a random forest classifier to predict the classification of a food item based on its nutrient composition. The model takes as input the log-transformed nutrient amounts per 100 grams of the selected food, and evaluates its likelihood of being classified according to the NOVA classification system. Specifically, FoodProX predicts whether a food falls into one of

four NOVA categories: 1 (unprocessed or minimally processed), 2 (processed culinary ingredients), 3 (processed foods), or 4 (ultra-processed foods).

To assess the effectiveness and consistency of FoodProX, we conducted a 5-fold stratified cross-validation on the labeled dataset in FNDDS 2009–2010. We computed the area under the receiver operating characteristic curve (AUC) and the area under the precision-recall curve (AUP) for 12, 62, and 99 nutrients, and the results were reported in Table S6. The reported metrics are the average and standard deviation over the 5 folds. In the main text, we present the overall performance by averaging the results across all three nutrient panels.

In order to enhance the classifier's ability to generalize to unseen data and mitigate over-fitting, we employed two strategies and retrained FoodProX accordingly. Firstly, we employed the Synthetic Minority Over-sampling Technique (SMOTE) to address the issue of class imbalance, which can often lead to biased predictions (this step is optional)[87]. Secondly, we created an ensemble voting system consisting of five classifiers, each trained on 4/5 of the generated dataset. The final predictions on unseen data are obtained by averaging the outputs of the five classifiers (see Box 1). By adopting these strategies, we aimed to improve the performance of FoodProX and ensure its robustness across different datasets.

The probability vector $\overrightarrow{p}_k = (p_1^k, p_2^k, p_3^k, p_4^k)$ is of particular importance as it reflects the level of confidence FoodProX has in assigning the four NOVA classes to a particular food item $k$. This vector belongs to the 4-D probability simplex, which comprises all vectors satisfying $\{\vec{p} \in \mathbb{R}^4, p_1 + p_2 + p_3 + p_4 = 1, \quad p_i \geq 0 \; \forall i\}$. The processing score $FPro_k$ defined in Eq. (1) is the orthogonal projection of $\overrightarrow{p}_k$ over the line going from the pure minimally processed state $\vec{p}_{MP} = (1,0,0,0)$ to the pure ultra-processed state $\vec{p}_{UP} = (0,0,0,1)$, represented by the explicit equation $p_1 = 1 - p_4$. Thanks to the ensemble strategy explained above and illustrated in Box 1, for each food we estimate the average $FPro$ and standard deviation.

For further details, see Section S2.

### Individual processing score *iFPro* and cross-sectional analysis

The individual Food Processing Score *iFPro* is a linear combination of the *FPro* scores of each food consumed by an individual, weighted according to the respective fraction of calories ($iFPro_{WC}$, Eq. (2)) or grams ($iFPro_{WG}$, Eq. S6) contributed to the diet. Although nutritional epidemiology research typically relies on a calorie-based score to assess dietary patterns, a weight-based index takes into account the consumption of highly processed beverages like zero-calorie soft drinks, which can have complex effects on health beyond their calorie content. Additionally, a weight-based index could also factor in food contaminants whose amount is independent of the number of calories provided, which can also impact overall health outcomes.

To investigate the relationship between *iFPro* and population health, we analyzed data from the NHANES 1999–2006 exposome and phenome database, which is a harmonized cross-sectional dataset created by Patel et al.[59]. This dataset consists of 255 publicly available data files from four cycles of NHANES, providing information on 41,474 individuals and 1191 variables. We focused on 20,047 adults (aged 18+) and calculated $iFPro_{WC}$ and $iFPro_{WG}$ using *FPro* estimates for 58 common nutrients across the selected NHANES cycles (see Table S4). We stratified the summary statistics for $iFPro_{WC}$ and $iFPro_{WG}$ by age, poverty income ratio, sex, race, and calories consumed, which are presented in Tables S7 and S8. To calculate the daily average *iFPro*, we used data from two-day dietary recalls obtained through in-person and phone interviews. For participants who did not complete two-day dietary recalls, we used data from in-person interviews only. We used survey weights to ensure the correct relevance of each individual for population statistics such as for the histograms shown in Fig. 3a–c[88,89].

For further details, see Section S3.

### Environment-wide association study

The environment-wide association study (EWAS) conducted on the merged NHANES 1999–2006 cohort, which is publicly available at ref. 59, aimed to identify and compare environmental factors and disease-related phenotypes that are strongly associated with $iFPro_{WC}$, $iFPro_{WG}$, and the fraction of calories contributed by manual NOVA 4. To achieve this goal, we gathered data on 45 exposure modules in ref. 59. Additionally, we included one variable to predict diabetes based on fasting glucose levels ≥126 mg/dL, as recommended by the American Diabetes Association[90], and two variables to predict metabolic syndrome[91]. We also included two assessments of the Framingham Risk Score[63,92] and the ACC/AHA Risk Score[64], which measure the 10-year risk of non-fatal myocardial infarction (MI), congestive heart disease (CHD) death, or fatal or non-fatal stroke.

To identify the most robust signal, we limited our analysis to variables that were measured in at least two cycles of NHANES (405 variables). We utilized survey-weighted generalized linear models to quantify the statistical associations, employing linear regression for continuous variables and logistic regression for categorical variables. All models were adjusted for age, sex, ethnicity, Body Mass Index (BMI), total-caloric intake, and estimated Socioeconomic Status (SES), as provided by NHANES and consistently with[90]. To account for the complex survey design of NHANES, we used the 'survey' statistical package in R[89]. Additionally, we filtered out categorical and continuous variables that did not meet a minimum sample size requirement for regression analysis. Specifically, we considered a ratio between the number of covariates and the number of data points ≤1/50 for continuous variables, and a similar threshold for the ratio between the number of covariates and the number of data points in the smallest category for categorical variables (Figure S17).

To improve the validity of our measures of association, we transformed all continuous variables using either the Box-Cox transformation or the logit function (in the case of the Framingham and ACC/AHA scores) to stabilize their variance[60]. We then standardized all continuous variables to place them on a similar scale. For multiple linear regression, we used fully standardized regression coefficients, which indicate the number of standard deviations of change in the dependent variable associated with one standard deviation increase in the independent variables. In logistic regression, we only partially standardized the continuous independent variables to maintain a straightforward interpretation of the relationship between one standard deviation increase in the Box-Cox transformed *iFPro* and the increase or decrease in disease odds[93].

To account for false discovery rate, we adjusted the $p$ values corresponding to each score using the Benjamini-Hochberg method with $\alpha = 0.05$. Our analysis revealed a total of 214 significant tests across the three methodologies, with 134 significant tests for $iFPro_{WC}$, 170 for $iFPro_{WG}$, and 92 for manual NOVA 4. The results of our analysis are summarized in Figures S17–S21. In addition, we compared our findings with literature results based on manual NOVA 4, which are presented in Table S9. The effect sizes calculated in EWAS were the inputs to the food substitution analysis implemented in Fig. 3f.

For further details, see Sections S4 and S5.

### Reporting summary

Further information on research design is available in the Nature Portfolio Reporting Summary linked to this article.

## Data availability

The data generated and analyzed in the study have been deposited on Zenodo at https://doi.org/10.5281/zenodo.7736993. A detailed source data file is provided with the manuscript. The publicly available datasets used in this study can be found on their associated websites: FNDDS (https://www.ars.usda.gov/northeast-area/beltsville-md-bhnrc/beltsville-human-nutrition-research-center/food-surveys-

research-group/docs/fndds-download-databases/), NHANES (https://www.cdc.gov/nchs/nhanes/index.htm), NHANES exposome and phenome data (https://github.com/chiragjp/nhanes_scidata), and Open Food Facts (https://world.openfoodfacts.org/data). Source data are provided with this paper.

## Code availability

The codes that support the findings of this study are openly available on our GitHub at https://doi.org/10.5281/zenodo.7736993 and https://github.com/menicgiulia/MLFoodProcessing. Python 3.6.10, MATLAB 2022a, and R (version 4.0.3 – 2020-10-10) were used for data analysis and visualization. The R Survey package (version 4.0) was used to incorporate survey weights in all analyses. No software was used for data collection.

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

## Acknowledgements

We thank Dr. Euridice Martinez Steele at the University of Sao Paulo for providing NOVA manual classification for FNDDS databases. We thank Dr. Wei Wang at Harvard Medical School for consulting as a statistical expert. This work was conducted with support from Harvard Catalyst | The Harvard Clinical and Translational Science Center (National Center for Advancing Translational Sciences, National Institutes of Health Award UL 1TR002541), and financial contributions from Harvard University and its affiliated academic healthcare centers. The content is solely the responsibility of the authors and does not necessarily represent the official views of Harvard Catalyst, Harvard University, its affiliated academic healthcare centers, or the National Institutes of Health. A.-L.B. is partially supported by NIH grant 1P01HL132825, American Heart Association grant 151708, and ERC grant 810115-DYNASET. D.M. is partially supported by the National Institutes of Health (2R01HL115189) and Vail Innovative Global Research (grant N316001 PR0677).

## Author contributions

G.M. and A.-L.B. conceived the project. G.M. designed the study, performed data modeling, analytical calculations, data query, and integration, and wrote the manuscript. B.R. performed data query, data integration, and statistical analysis, and contributed to writing the manuscript. D.M. contributed to interpreting the results and to writing the manuscript. A.-L.B. contributed to the conceptual design of the study and to writing the manuscript.

## Competing interests

A.-L.B. is the founder of Scipher Medicine and Naring Health, companies that explore the use of network-based tools in health and food. D.m. reports research funding from the National Institutes of Health, the Gates Foundation, the Rockefeller Foundation, Vail Innovative Global Research, and the Kaiser Permanente Fund; personal fees from Acasti Pharma and Barilla; scientific advisory board of Beren Therapeutics, Brightseed, Calibrate, Elysium Health, Filtricine, HumanCo, Instacart Health, January Inc., and Perfect Day (ended: Day Two, Discern Dx, Season Health, and Tiny Organics); stock ownership in Calibrate and HumanCo; and chapter royalties from UpToDate, all outside the submitted work. The remaining authors declare no competing interests.
