## [Peer Review File · Nature Communications]

Machine Learning Prediction of the Degree of Food ProcessingREVIEWER COMMENTS

Reviewer #1 (Remarks to the Author):

This is a creative paper to create a processed food score, called FProX to automate the classification of processed food. Currently, the nutritional field is missing ways to systemically assess the nutritional content of groups of foods as current methods lack the scale that is claimed by the current approach.

Several comments and critiques:

1. How generalizable is the approach to cohort data outside of the NHANES?
2. How reproducible are the existing NOVA criteria?
3. The EWAS is a nice addition; however, what are the anticipated longitudinal associations between processed food index and disease?
4. How interpretable are the associations versus doing an independent association between a processed food and ProX - compare the R2 of the two approaches (highly processed food by Pro 1 or 4 vs IProX or French fries vs. IProX)
5. IProX seems poised for population screening, but difficult to ascertain causal connections between processed foods, where specificity is required, and a biological outcome (As is attempted to be discussed in the Discussion, under "Food Substitution") . For example, if two groups of foods had the same IProX score, would the outcomes be the same? Different? What are the implications of each scenario?

Reviewer #2 (Remarks to the Author):

Thank you for this manuscript. I found the topic highly relevant and important. The need for an algorithm to classify processed foods into different types of processing is justified. However, I see that authors are making several unfounded claims about NOVA to justify their tool. My reading is that authors did not fully understand the NOVA system in the way it is described and no definitions of ultra-processed food is given, therefore it is hard to understand if authors are using a similar definition of ultra-processed to the one proposed by NOVA or another. My understanding is that the paper proposed a nutrient-based definition of ultra-processed food.

There are several adjectives and claims made in the paper about NOVA that are not accurate and authors should address those in the revision of the paper. I have highlighted these passages in yellow in the attached paper.

For example, NOVA does not consider all ultra-processed foods to be identical, and it provides an operational definition of ultra-processing which authors have omitted to report in their paper. Instead, authors claimed that ultra-processed foods are all the rest of foods not classified as group 1, 2 and 3,

which is not correct. NOVA does classify all food as ultra-processed when there is a cosmetic additive or a substance of non-culinary use in the ingredient list. This is the basis of the definition and although it can be criticized of course it is not a limitation but part of the definition.

Authors are right saying that the NOVA system is laborious to apply (because it requires expert judgement and is manual) and requires the list of ingredients which is not always available, and this seldom justifies the need for an algorithm, but not because NOVA is incomplete. NOVA allow for a more refined analysis by studying how subgroups of ultra-processed food are related or not to health outcomes (ex: soft drinks vs sweetened yogurts) but so far, studies have examined the ultra-processed group as a whole, which show consisting results, but nevertheless NOVA allow for a more refined examination of the association between types of ultra-processed food, diet and health. I think this is an important nuance to make.

NOVA defines four groups based on the degree and purpose of processing, not just the degree as authors claimed. 1

The statement that NOVA covers only 35% of the USDA database is unclear to me, since there are several studies based on NHANES examining ultra-processed food, diet and health. Please explain.

Line 103 : define AUC

Lines 118-120 : Here, it is clear to me that authors are working on a different conception of foods which is nutrient-based. NOVA is not a nutrient-based classification, it does not only consider the degree of processing that alter nutrient content, but also the purpose of processing, which is I understand a harder concept to quantify. Therefore, the claim that the ambiguity found suggest that a four-based classification does not capture nutrient profile accurately is incorrect, because NOVA is not intended to classify food on a nutrient basis, but more on the type of ingredients which reflects the purpose of processing. This statement need to be revised. In NOVA, 'onions sautees' would be disaggregated into its main ingredients as Group 1 (raw onion), Group 2 (oil, salt), because NOVA is intended to classify foods the ways they are bought. An 'onion sautés' prepared by industry and sold in stores would be Group 3 (raw onion+oil+salt), because it is bought as ready to eat. NOVA thus makes the distinction between what is prepared in kitchen and was is bought from stores. Although one can of course criticized this approach, it needs to be proper understood to be criticized and challenged.

Lastly, substitution of foods, as described, may work in theory, but it does imply a change of culture and behavior around foods which is more complex that the model suggests. However, studies on NOVA have shown that the more ultra-processed food is consumed, the less fresh/unprocessed food are consumed, and also ultra—processed food have addictive properties, and these properties are partly due to the design of ultra-processed food (use of additives marketing, branding, etc.) which show that the purpose of processing matters beside nutrient content and degree of processing.

Lines 328-330: The statement need revision. I don't understand the sentence, if 70% of foods are ultra-processed (Or 73% in your study), then how your model is more feasible that the one suggested by NOVA? NOVA suggest to favor fresh and minimally processed food, and to reduce ultra-processed food. Your model suggests to replace UPF with less processed version, this imply also several challenges. I think that both approaches have the same type of limitations: cost, availability, taste preferences, culinary skills, time, etc.

Line 332: here it is clear that the model is nutrient-based. This may be more effective that NOVA, but it needs to be explained because as the paper is now, the algorithm and NOVA are not based on the same approach, therefore I am not sure if authors can claim that they are classifying ultra-processed food! They may be classifying instead highly-processed food (based on a nutrient classification), and this should be made clear.

Reviewer #3 (Remarks to the Author):

The authors proposed a method for classifying food products based on their nutrient profile into four classes of processing food levels (NOVA system). Even the problem is worth to be investigated, I think that there is much more to be done in order this paper to be published.

Main questions:

1. The presented results are weak. NOVA system is only one system that can be used to classify the level of processed foods. FoodEx2 language is a language published by the European Food Safety Authority that also consists of four classes based on the level of processing. Even more, they also support facet terms where the food process can be described on a level of raw, derivative, simple and composite food. LanguaL also support this kind of information. The main problem to make this automatically with machine learning is missing annotated data with regard to this systems or the other systems that exist. This is completely not mention in the manuscript.
2. The model is a simple random forest model for multi-class classification trained on an annotated dataset based on nutrient composition of the products. However, from practical point of view, this can be challenging since there are a lot of missing values in the food composition databases that limit the application in practice.
3. Going from 16 to 12-gram nutrient is only a feature selection problem in ML.
4. There are previous work, where instead of RF model, an ensemble of classifiers were used which

based on the textual description of the food can classify different levels of processing based on FoodEx2 system.

5. Page 4. "Have significant predictive power" - From where this can be concluded?

6. Details about learning the model, validation of the model, selecting its hyper-parameters are not mentioned.

7. What is the distribution of the classes in the training data and testing data? Is this was repeated?

8. Using the PCA cannot lead to explainable results. SHAP values can be used for more interpretable results and providing insight in the decisions made by the model.

9. If the classes assigned to unlabelled products were validated by domain experts?

10. The model cannot be reproduce, details are missing, data is also not available.

Reviewer #4 (Remarks to the Author):

Review, Mike Rayner, University of Oxford, 12th August 2022

General points

This paper introduces 'FoodProX, a machine learning classifier trained to predict the degree of processing of any food. Importantly, FoodProX allows {the authors} to define a continuous index that captures the degree of processing of any food, and can help quantify the overall diet quality of individuals, unveiling statistical correlations between the degree of processing characterizing individual diets and multiple disease phenotypes.' (Line 65) It claims that 'The remarkable ability of the FoodProX predictions to replicate the manual NOVA classification confirms that food processing results in distinct patterns of nutrient alterations, accurately detected by machine learning. (Line 315)

This is really interesting and useful paper that will make a major contribution to debates about the impact of food processing on health and on the usefulness of food profiling systems that are entirely or even partially based on the degree of processing of a food. The paper, in effect, proposes an alternative (FoodProX) to the NOVA classification system – the best known and most extensively studied food scoring systems based on the degree of processing of a food up until now.

FoodProX differs from NOVA in generating a continuous score for the degree of processing of a food whereas NOVA classifies foods into four groups: unprocessed food, culinary ingredients, processed foods, ultra-processed food. FoodProX is based on the nutrient content of a food whereas NOVA relies on a person (or persons) manually allocating a food to one of the four groups based on a large number of considerations. For example, as I understand it, a chocolate cake made in the home would not be considered an ultra-processed food NOVA whereas a chocolate cake made in a factory would. As the authors suggest this means that FoodProX has considerable advantages over NOVA.

My only reservation about this paper is there seems to be some circularity in the arguments. In particular since FoodProX was developed using a training dataset that had foods classified according to NOVA, the ability of the FoodProX predictions to replicate the manual NOVA classification is hardly 'remarkable' whereas Line 315 suggests that it is.

I am also not convinced that the ability of FoodProX to replicate NOVA does 'confirm that food processing results in distinct patterns of nutrient alterations, accurately detected by machine learning. (Line 315 again). This conclusion suggests a causal relationship between processing (as defined by NOVA) and changes in nutrient levels. Of course this is likely to be the case, as demonstrated by the authors' persuasive example of the changes in nutrients between uncooked onion and processed onion. But this causal relationship is not, I think, demonstrated by the ability of FoodProX to replicate NOVA classifications.

I think the authors mean something like: 'The classifications of the NOVA system are associated with distinct patterns of nutrients that can be detected by machine learning'.

However, on balance, I think this paper is really interesting in both its main findings and some of its 'incidental' finding such as 'Currently, our analysis in Section S1.5 shows that an unsupervised hierarchical clustering of foods, leveraging the widest nutrient panel available in FNDDS, can [sic] not able to independently reproduce the four NOVA classes'.

I think the Discussion could be longer but I also rather like its brevity which leaves the reader the freedom to draw their own conclusions.

Specific comments.

Line 46 'First, as the current categorization relies on expertise-based manual evaluation of each food, it covers only 35% of the foods catalogued by U.S. Department of Agriculture (USDA) (SI Section 1).' I recommend making this sentence clearer. What current categorisation are we talking about here? Presumably, the authors mean the NOVA classification system?

Line 188 'In other words, while information on additives can improve FProX's performance, changes in

the nutrient profile already carry the bulk of the predictive power.’ This is an interesting observation but it begs the question of the extent to which information on changes in the nutrient profile would improve an algorithm based on additives. And also whether it is the additives or the nutrient changes which make processed foods ‘unhealthy’. Perhaps this could be discussed in the Discussion.

Line 315 ‘The remarkable ability of the FoodProX predictions to replicate the manual NOVA classification confirms that food processing results in distinct patterns of nutrient alterations, accurately detected by machine learning.’ I recommend deleting the word ‘remarkable’ and changing second half of sentence as discussed above.

Line S452 ‘Currently, our analysis in Section S1.5 shows that an unsupervised hierarchical clustering of foods, leveraging the widest nutrient panel available in FNDDS, can [sic] not able to independently reproduce the four NOVA classes.’ Change ‘can’ to ‘is’.

Reviewer #1 (Remarks to the Author):

This is a creative paper to create a processed food score, called FProX to automate the classification of processed food. Currently, the nutritional field is missing ways to systemically assess the nutritional content of groups of foods as current methods lack the scale that is claimed by the current approach.

We thank the Referee for the constructive recommendations, and for finding our paper creative and timely. In the following, we address point-by-point all the questions and recommendations offered by the Reviewer. We also modified the main text and SI to address each of the Referee's suggestions. We hope the Referee will find the revised manuscript appropriate for publication.

Several comments and critiques:

1. How generalizable is the approach to cohort data outside of the NHANES?

We thank the Referee for giving us the chance to elaborate on the generalizability of our approach beyond NHANES. Our algorithm is designed to work with a minimal number of nutritional measures, making it adaptable and portable to other food composition databases. This adaptability enabled us to analyze different cycles of NHANES, with nutrient panels of varying size (from 58 to 99), and then to extend our analysis to over 50,000 products collected from major grocery store websites [1], and to 233,831 food products worldwide in Open Food Facts (see Results Section on “Stability and Robustness of FPro”), for which only nutrition facts are available.

2. How reproducible are the existing NOVA criteria?

Thank you for raising this very important point. Nutritional epidemiologists are increasingly using NOVA to explore relationships between the consumption of highly processed foods and diet quality or health outcomes. Indeed, NOVA was used in 95% of the studies on this topic published between 2015 and 2019, and which have been included in a recent systematic review [2]. However, because NOVA classification approach is purely descriptive in nature, it leaves room for ambiguity and differences in interpretation [3]. This is exactly the reason why we need an algorithm like FPro that takes a standardized input and provides reproducible results.

To remove any ambiguity in NOVA class interpretation, our algorithm was trained only on the foods manually labeled by Prof. Monteiro's group, the creator of NOVA.

Prompted by the Reviewer's question, we rephrased Lines 36-85, and Lines 109-1110.

3. The EWAS is a nice addition; however, what are the anticipated longitudinal associations between processed food index and disease?

Thank you for appreciating the EWAS pipeline. The anticipated longitudinal associations are with cancer, cardiovascular disease, depressive symptoms, visceral fat, overweight, obesity, and type 2 diabetes, which are in line with earlier findings regarding the potential impact of ultra-processed food consumption by longitudinal cohorts such as NutriNet Sante' [4]–[9], PREDIMED [10], Elsa-Brasil [11], Health Professionals' Follow-up Study and Nurses' Health Study [12]. These associations were found significant with the standard NOVA classification,

and given the higher predictive power of FPro and our current findings in publicly available cross-sectional cohorts, we are confident to replicate them, as well as unveil new signals and correct for spurious results (see Lines 304-311 and SI Section 4, SI Figures 18, 19, 20 and 21). In Table S9 we provide a summary of the epidemiological literature on the discovered associations with Manual NOVA 4, which are then explicitly compared to our findings in NHANES.

4. How interpretable are the associations versus doing an independent association between a processed food and ProX - compare the R2 of the two approaches (highly processed food by Pro 1 or 4 vs IProX or French fries vs. IProX)

We fully agree with the Reviewer on the importance of interpretable scores and inspired by the Reviewer's question we added SI Section 3.5, where we investigated the relation between iFPro population values and the fraction of calories consumed in each What We Eat in America (WWEIA) food category. First, we calculated Spearman's rank correlation between iFPro and the individual caloric consumption stratified over 151 WWEIA food classes (Figure S15). Second, we compared the caloric consumption stratified by food groups in the first quintile of iFPro (Q1), i.e., individuals with the least ultra-processed diet, with the last quintile of iFPro (Q5), i.e., individuals with the most-ultra processed diet. Differences in caloric consumption were identified through the Mann-Whitney U rank test, for which we calculated p-value and effect-size (Figure S16). All the results were corrected for multiple testing.

For instance, "French fries and other fried white potatoes" is the third most positively correlated food group with iFPro ($r=0.18$), after "Soft drinks" ($r=0.36$), and "Burgers" ($r=0.21$). Indeed, individuals in Q5 consume on average 3.55 times more French fries than Q1. On the other hand, the consumption of "White potatoes, baked or boiled" is negatively correlated with iFPro ($r=-0.12$), as individuals in Q1 consume an average of 3.89 times more baked potatoes than Q5.

5. IProX seems poised for population screening, but difficult to ascertain causal connections between processed foods, where specificity is required, and a biological outcome (As is attempted to be discussed in the Discussion, under "Food Substitution") . For example, if two groups of foods had the same IProX score, would the outcomes be the same? Different? What are the implications of each scenario?

If two foods have the same FPro, they do not lead to the same decrease in iFPro, as their substitution in each individual diet is prioritized and weighted according to the fraction of calories each food contributes to an individual's diet, and by the availability of an alternative food with significantly smaller FPro in the respective food category.

In other words, each food in an individual's diet is first classified according to WWEIA and then assigned to a measure of relevance (Eq. S8) for the substitution algorithm: foods that are selected first contribute to a high fraction of calories and have within the same category an alternative with significantly smaller FPro. In this scenario, individuals keep consuming the same food categories, and the FPro of each food is compared only to other foods within the same group, for which a similar FPro implies a very close nutrient profile. According to this metric, finding two foods with exactly the same relevance for the algorithm is nearly impossible. Regarding food groups, as shown in Figure 2D, they are not characterized by a single FPro, but by distributions of FPro values.

In summary, we wish to thank the Reviewer for prompting us to add a very interpretability analysis to the paper, which has undoubtedly improved the quality of our manuscript, and for the many constructive observations on the manuscript.

Reviewer #2 (Remarks to the Author):

Thank you for this manuscript. I found the topic highly relevant and important. The need for an algorithm to classify processed foods into different types of processing is justified. However, I see that authors are making several unfounded claims about NOVA to justify their tool.

We thank the Referee for finding this paper timely and highly relevant. In the following, we address all the questions and recommendations offered by the Reviewer. We hope that the revised manuscript will clarify any potential misunderstanding our previous formulation may have led to.

1) My reading is that authors did not fully understand the NOVA system in the way it is described and no definitions of ultra-processed food is given, therefore it is hard to understand if authors are using a similar definition of ultra-processed to the one proposed by NOVA or another. My understanding is that the paper proposed a nutrient-based definition of ultra-processed food.

We thank the Reviewer for prompting us to clarify the relation between our work and NOVA classification. Our goal is to create a data-driven measure of the degree of food processing, able to work with a minimal number of nutritional measures, making it reproducible and portable to different food systems and cohort studies. In other words, we are not offering a nutrient-based definition of ultra-processed food. Rather, our algorithm learns from the existing NOVA manual classification to identify the NOVA class from nutrient data only.

Why nutrients? 1) they are consistently regulated and reported worldwide, 2) their ranges in unprocessed food are constrained by physiological ranges determined by biochemistry [13], 3) food processing systematically and reproducibly alters their concentrations (Figure 1A -B), making it an ideal application for machine learning.

To teach our algorithm how to score processing from nutrients we decided to leverage NOVA, the most used system to classify foods according to processing-related criteria [2]. This choice provides us with a wealth of epidemiological literature to compare with.

To remove any ambiguity due to NOVA class interpretation [3], our algorithm was trained only on the foods classified by Prof. Monteiro's group, the creator of NOVA. This implies that our work is in line with the NOVA class definitions provided in [14], [15], and in all the sub-sequent papers from the same group.

The definition of NOVA classes is provided at Lines 36-45. We have now reformulated the paragraphs to make them more precise, as the Reviewer suggested. Full details about the NOVA labels shared by Prof. Monteiro's group are available in SI Section 1.2 and 1.3.

2) There are several adjectives and claims made in the paper about NOVA that are not accurate and authors should address those in the revision of the paper. I have highlighted these passages in yellow in the attached paper. For example, NOVA does not consider all ultra-processed foods to be identical, and it provides an operational definition of ultra-processing which authors have omitted to report in their paper. Instead, authors claimed that ultra-processed foods are all the rest of foods not classified as group 1, 2 and 3, which is not correct. NOVA does classify all food as ultra-processed when there is a cosmetic additive or a substance of non-culinary use in the

ingredient list. This is the basis of the definition and although it can be criticized of course it is not a limitation but part of the definition.

We apologize for the oversimplification used in the previous formulation of our manuscript. Our intention was to reflect how NOVA classification is currently used by nutrition experts [2], [3]. We have removed “everything else”, and reformulated Lines 36-45 using Prof. Monteiro’s definitions [14], [15].

3) Authors are right saying that the NOVA system is laborious to apply (because it requires expert judgement and is manual) and requires the list of ingredients which is not always available, and this seldom justifies the need for an algorithm, but not because NOVA is incomplete. NOVA allow for a more refined analysis by studying how subgroups of ultra-processed food are related or not to health outcomes (ex: soft drinks vs sweetened yogurts) but so far, studies have examined the ultra-processed group as a whole, which show consisting results, but nevertheless NOVA allow for a more refined examination of the association between types of ultra-processed food, diet and health. I think this is an important nuance to make.

The Reviewer is correct — our goal was to reflect how NOVA is used in practice by the scientific community [2], [3], [16] and to maximize our ability to compare our results with the epidemiological results on ultra-processed food (see Results-Health Implications and SI Section 4). We chose to work with four NOVA classes to be consistent with the labels provided to us by Prof. Monteiro’s group [14], [15] (see Acknowledgements) and his general advice to substitute NOVA 4 products with NOVA 1 and NOVA 3 foods [17]. However, we agree with the Reviewer, therefore we reformulated the sentence at Lines 70-71 to indicate to the reader that NOVA allows for a more refined analysis.

4) The statement that NOVA covers only 35% of the USDA database is unclear to me, since there are several studies based on NHANES examining ultra-processed food, diet and health. Please explain.

We thank the Reviewer for prompting us to clarify what we mean by coverage, as the concept has indeed important nuances. While FNDDS (the USDA food composition database connected to NHANES 24hrs recalls) was thoroughly investigated by NOVA researchers, only 35% of the database has a unique NOVA label per item. The reason is that NOVA classification is not available for composite dishes unfolded into the underlying ingredients, which represent 65% of the food items. We now clarify this aspect at Lines 52-63 in the manuscript. A detailed description of Prof. Monteiro’s group work on NHANES is provided in SI Section 1.3, where in Figure S3 we show the percentage of FNDDS food not classified (~10%) and the percentage of FNDDS foods that needs further decomposition in ingredients to be correctly assessed (~60%), based on the data shared by Dr. Martinez Steele.

5) Line 103 : define AUC

The Area Under the Receiver Operating Characteristics (AUC) quantifies the ability of a classifier to distinguish between different classes. In more strictly mathematical terms, AUC is defined as $AUC = P(X_1 > X_0)$, where X_1 is a continuous random variable describing the output

of a binary classifier for a randomly chosen positive sample, and X_0 is a continuous random variable describing the output of the same classifier for a randomly chosen negative sample. Therefore, AUC measures the probability that a classifier gives a higher score to a positive sample (the class of interest), compared to a negative one (everything else).

In this manuscript, we apply AUC to measure how well FoodProX discriminates between different NOVA classes, leveraging only the nutritional information. For instance, an AUC of 0.9804 for NOVA1 means that FoodProX successfully distinguishes a true NOVA1 food from a randomly picked item from other NOVA classes 98.04% of the times. AUC is a standard measure of performance in Machine Learning: the best performances are close to 1, while a random classifier with no class separation capacity has AUC=0.5.

Prompted by the Reviewer's question, we added the definition of AUC at Lines 119-122. Furthermore, in Section 2.1 we provide a detailed characterization of the performance of FoodProX.

6) Lines 118-120 : Here, it is clear to me that authors are working on a different conception of foods which is nutrient-based. NOVA is not a nutrient-based classification, it does not only consider the degree of processing that alter nutrient content, but also the purpose of processing, which is I understand a harder concept to quantify. Therefore, the claim that the ambiguity found suggest that a four-based classification does not capture nutrient profile accurately is incorrect, because NOVA is not intended to classify food on a nutrient basis, but more on the type of ingredients which reflects the purpose of processing. This statement need to be revised

We thank the Referee for raising this important point and giving us the chance to fully clarify the aims of our manuscript. We are aware of NOVA's philosophy, thanks to our repeated and ongoing discussion with Prof. Monteiro's group. Indeed, in the first lines of Monteiro et al. [14] we read: "NOVA is the food classification that categorizes foods according to the extent and purpose of food processing, rather than in terms of nutrients." Our ultimate goal was not to predict NOVA classes, but to create a score that is able to reproducibly rank foods within the *same food category*, hence to help us optimize our intervention strategies. This goal differs from NOVA's, which does not aim to capture the degree of processing of any food preparation technique, but rather to group foods based on the degree of industrial processing. To validate our hypothesis that food processing alters in a reproducible way the natural ranges of nutrient amounts in unprocessed food, we needed to test the predictive power of nutrient concentrations against the widely used NOVA system. Interestingly, while NOVA researchers do not use nutrient concentrations to classify food, our algorithm, leveraging nutrients as input, does predict the known NOVA labels with extremely good precision (Lines 122-126), suggesting that nutrients mirror the bulk of information regarding food processing. The ability to predict NOVA is then a validation, a means to an end rather than the goal of our project.

In summary, we agree with the Referee on the need to better clarify the goals of our manuscript. To that end, following the Reviewer's suggestions, we revised Lines 140-142 and added Lines 344-349 to the Discussion.

7) In NOVA, 'onions sautees' would be disaggregated into its main ingredients as Group 1 (raw onion), Group 2 (oil, salt), because NOVA is intended to classify foods the ways they are bought. An 'onion sautés' prepared by industry and sold in stores would be Group 3 (raw

onion+oil+salt), because it is bought as ready to eat. NOVA thus makes the distinction between what is prepared in kitchen and what is bought from stores. Although one can criticize this approach, it needs to be properly understood to be criticized and challenged.

We thank the Referee for bringing to our attention a clear example of how NOVA classification is open to different interpretations, as discussed in [3]. Given our access to the data annotated by Monteiro's group, we are able to provide the exact details of their classification process. In the manual classification, 'Onions, mature, cooked or sauteed, from fresh, fat added in cooking' is unfolded into 'Onions, cooked, boiled, drained, without salt', 'Margarine-like spread, tub, salted' and 'Salt, table', and each of these ingredients is classified according to NOVA (group 1, 4 and 2, respectively). This implies that 'onion fried/sauteed' does not have a unique NOVA class. Furthermore, the level of details available in FNDDS and NHANES allows for an additional level of evaluation: if the source of food is known (see SI Section 2.6) and for instance, 'onion fried/sauteed' was consumed at a 'Restaurant fast food', the food item is no further decomposed into ingredients and it is assigned to NOVA 4.

Overall, the probability values for 'onion fried/sauteed' found by FoodProX are:

p ₁	p ₂	p ₃	p ₄
0.096278	0.0027778	0.24883	0.65211

suggesting an intermediate level of processing, one that is better captured by a continuous score such as FPro rather than by the distinct NOVA classes.

8) Lastly, substitution of foods, as described, may work in theory, but it does imply a change of culture and behavior around foods which is more complex than the model suggests. However, studies on NOVA have shown that the more ultra-processed food is consumed, the less fresh/unprocessed food are consumed, and also ultra-processed food have addictive properties, and these properties are partly due to the design of ultra-processed food (use of additives marketing, branding, etc.) which show that the purpose of processing matters beside nutrient content and degree of processing.

We agree with that the Reviewer that our substitution strategy does not take into account many behavioral aspects and addictive properties of ultra-processed food, currently not captured by the available data. Empowering consumers with information on the degree of processing characterizing the foods they purchase is an important first step towards reducing the reliance on more highly processed food. We agree with the Referee that many additional factors have to be considered in order to make this strategy successful. In that spirit, we crafted a minimal intervention strategy, preserving the general nature of an individual's diet, as we looked into the challenges of altering diet and eating behaviors [18], [19].

This is an excellent point made by the Reviewer and we realized that this aspect has not been spelled out in the previous version of the manuscript — so we now clarified it at Lines 335-342.

9) Lines 328-330: The statement needs revision. I don't understand the sentence, if 70% of foods are ultra-processed (or 73% in your study), then how your model is more feasible than the one suggested by NOVA? NOVA suggests to favor fresh and minimally processed food, and to reduce ultra-processed food. Your model suggests to replace UPF with less processed version, this

imply also several challenges. I think that both approaches have the same type of limitations: cost, availability, taste preferences, culinary skills, time, etc.

We are happy to clarify this sentence. As the Reviewer mentioned, nutritional guidelines based on NOVA suggest consuming “in natura” and minimally processed food, limiting processed food (NOVA 3), and avoiding ultra-processed food (NOVA 4) [17], [20], [21]. However, we estimated that approximately 73% of the US food supply is NOVA 4, in agreement with other independent analyses such as [22]. Considering the challenges of altering diet and eating behaviors [18], [19], for the reasons the Referee mentions (cost, availability, taste preferences, culinary skills, time), it is impractical to advise people to shift their dietary patterns towards the remaining 30% of the food supply. By preserving the general nature of an individual’s diet and relying on a continuous processing score, we are able to offer recommendation within the 70% of the food supply, nudging people towards progressively less processed diets.

We agree with the Reviewer on the presence of limitations for both approaches, and following the Reviewer’s suggestions, we now acknowledge the role of major disparities in life-style and healthy food access at Lines 335-342.

10) Line 332: here it is clear that the model is nutrient-based. This may be more effective than NOVA, but it needs to be explained because as the paper is now, the algorithm and NOVA are not based on the same approach, therefore I am not sure if authors can claim that they are classifying ultra-processed food! They may be classifying instead highly-processed food (based on a nutrient classification), and this should be made clear.

Prompted by the Referee’s recommendation, we now clearly state that our model combines features of processing techniques learned from manual NOVA labels, with nutrient concentrations from food composition data (Lines 344-349). The goal of FPro is not to classify ultra-processed food from NOVA’s perspective, but to create a continuous score capturing the chemical-physical alterations of food compared to the expected nutrient values in raw ingredients, driven by various processing techniques.

In summary, we wish to thank the Reviewer for prompting us to clarify many aspects of our work, which has undoubtedly improved the quality of our manuscript, making it more accessible and broadening its impact.

Reviewer #3 (Remarks to the Author):

The authors proposed a method for classifying food products based on their nutrient profile into four classes of processing food levels (NOVA system). Even the problem is worth to be investigated, I think that there is much more to be done in order this paper to be published.

We thank the Reviewer for finding the object of this paper of potential interest. In the following, we address all the suggestions and questions raised by the Reviewer. We hope that the revised manuscript will clarify any potential misunderstanding our previous formulation may have led to.

Main questions:

1. The presented results are weak. NOVA system is only one system that can be used to classify the level of processed foods. FoodEx2 language is a language published by the European Food Safety Authority that also consists of four classes based on the level of processing. Even more, they also support facet terms where the food process can be described on a level of raw, derivative, simple and composite food. LanguaL also support this kind of information. The main problem to make this automatically with machine learning is missing annotated data with regard to this systems or the other systems that exist. This is completely not mention in the manuscript.

The Reviewer is correct in stating that other systems address food processing. Following the Reviewer's suggestions we now explicitly mention and discuss them at Lines 32-35 in the introduction.

To teach our algorithm how to measure food processing from nutrient concentrations we decided to leverage NOVA classification, which is currently the most widely used, hence it offered us a wealth of epidemiological literature to compare our analysis with (see Results-Health Implications and SI Section 3 and 4). Indeed, nutritional epidemiologists are increasingly using NOVA to explore relationships between the consumption of highly processed foods and diet quality or health outcomes: NOVA was used in 95% of the studies on this topic published between 2015 and 2019, and which have been included in a recent systematic review [2]. Furthermore, policymakers are moving to use NOVA assignments to guide national and international public health decisions [21], [23]. For example, several Latin America countries have constructed dietary guidelines based on using NOVA [17], [20], and the French government is drawing upon NOVA in its objective to reduce ultra-processed food consumption by 20% [24].

The Reviewer also made an excellent point regarding the lack of automatization due to missing annotated data. Indeed, we were successful in training and testing our algorithm because Prof. Monteiro's group, the creators of NOVA and the first group defining the concept of "ultra-processed food", provided us with manually annotated food composition data. For further details see Lines 109-110 and SI Section 1.

2. The model is a simple random forest model for multi-class classification trained on an annotated dataset based on nutrient composition of the products. However, from practical point of view, this can be challenging since there are a lot of missing values in the food composition databases that limit the application in practice.

We agree with the Reviewer on the issue of missing values in food composition databases. This is precisely the reason why we trained and tested on FNDDS, a USDA food composition database carefully designed to *have no missing values*. Indeed, FNDDS is designed by the USDA to provide food composition data for foods and beverages reported in the dietary component of the National Health and Nutrition Examination Survey (NHANES), a biannual cross-sectional survey of the US Population conducted by Center for Disease Control and Prevention (CDC) to monitor the health of Americans. FNDDS is derived by combining the food items provided in the USDA National Nutrient Database for Standard Reference (SR). In other words, each item in FNDDS is related to one or more foods in SR, reported as ingredients in FNDDS. Differently from SR, designed for the dissemination of food composition data, FNDDS's goal is to enable the analysis of dietary intake, hence it contains no missing nutrient values, hence offering an ideal database to train machine learning models. A detailed description of FNDDS is available in SI Section 1.2.

Using FNDDS we managed to train and test models leveraging different subsets of nutrients, as mentioned at Lines 109-126, and documented in SI Section 2.1. Of particular importance for application purposes, is our model based on the 12 nutrients provided by the nutrition facts, information strictly regulated and mandated by law on food packaging, hence once again complete, with no missing values. This model allowed us to extend our analysis to over 50,000 products collected from US major grocery store websites [1], and to 233,831 food products worldwide in Open Food Facts (see Results Section on “Stability and Robustness of FPro”), for which only nutrition facts are available.

3. Going from 16 to 12-gram nutrient is only a feature selection problem in ML.

At Line 118 “12 gram-based nutrients” refers to the input to one of our models, built to classify and score all products reporting the 12-gram-based nutrients mandated by FDA. Using FNDDS we managed to train and test models leveraging different subsets of nutrients, as mentioned at Lines 109-126, and fully documented in SI Section 2.1.

4. There are previous work, where instead of RF model, an ensemble of classifiers were used which based on the textual description of the food can classify different levels of processing based on FoodEx2 system.

We thank the Reviewer for pointing us to the work on FoodEx2. We are now citing the related papers at Line 35. Unfortunately, in all databases we analyzed, the food description was limited and/or lacked standardization, leading to a document-term matrix (derived from lemmatization or stemming of the text available in the manually labeled database) poorly generalizable to new food databases. Similar considerations stand for word embeddings trained on food-related corpus. This is the reason why we focused on nutrient content, as it is consistently regulated and reported worldwide. In [1], for branded food products, we started exploring through text analysis

and NLP the predictive power of ingredient lists, since they are more strictly regulated by the FDA, compared to food names and descriptions.

5. Page 4. "Have significant predictive power" - From where this can be concluded?

Thank you for prompting us to clarify this aspect. In this case, we are comparing the performance of FoodProX with a random classifier. We have reformulated Lines 124-126 to make this passage clearer. The Reviewer can access a detailed analysis of the ROC curves and Precision-Recall curves in SI Section 2.1, over a 5-fold stratified cross-validation.

6. Details about learning the model, validation of the model, selecting its hyper-parameters are not mentioned.

Following the Reviewer's suggestion we added the details on the selection of the hyper-parameters through a randomized search with stratified cross-validation at the beginning of SI Section 2.1, Lines 146-154. SI Section 2, from page 10 to page 19 is dedicated to the performance of the algorithm and its validation with case-studies. In the manuscript, in Lines 172-226 we discuss the stability and robustness of our predictions.

7. What is the distribution of the classes in the training data and testing data? Is this was repeated?

Happy to clarify this. In Figure SI 2C we show the class distribution of the training data set. Of the 2,971 labeled items, 11.4% were NOVA 1, 1.8% NOVA 2, 15.7% NOVA 3, and 71.1% NOVA 4. The algorithm was trained and tested with a stratified 5-fold cross-validation (SI Section 2.1).

8. Using the PCA cannot lead to explainable results. SHAP values can be used for more interpretable results and providing insight in the decisions made by the model.

Thank you for raising this very important point and giving us the chance to further elaborate on this. The PCA is not used in the analysis for model explainability but only for visualization purposes, to embed the 4D probability space of the classifier in 2D and explain geometrically the relation with FPro. At Lines 129-130 in the manuscript, we point to the extensive feature importance analysis available in SI Section 2.2.

Prompted by the Reviewer's suggestion, we extended SI Section 2.2 adding a SHAP analysis at Lines 193-221. Overall, the results are consistent with our previous analysis leveraging permutation feature importance [25].

9. If the classes assigned to unlabelled products were validated by domain experts?

FPro is based on FoodProX that has been extensively validated with a 5-fold stratified cross-validation against manual domain-expert NOVA labels in FNDDS (Table S6). We observe similar performances on the annotations provided by Open Food Facts, indicating that FoodProX is stable and consistent across databases (Table S10). However, these scenarios describe labeled data. If foods in the studied databases were unlabeled, they were likely decomposed into

ingredients to be classified, as we describe in SI Section 1.3. This reflects the challenges of applying NOVA classification to composite food, as discussed in the manuscript at Lines 52-63. From a machine learning perspective, the classification problem is then not “closed”, as the labels 1, 2, 3, 4 do not univocally and exhaustively cover all foods. This is the reason why we devoted the majority of our efforts to create and validate the continuous processing score FPro. To do so, we involved as co-author Dr. Mozaffarian, a domain-expert, dean at the Tufts Friedman School of Nutrition Science and Policy, and leader of the White House Conference on Hunger, Nutrition, and Health. He provided a nutritional perspective and validation of FPro, and suggested systematic tests for our methodology, such as the case study on onion shown in the manuscript, the case study on industrial cereals and their ingredients discussed in Section S2.5, and the analysis of the relation between FPro and number of additives in products discussed in Section S6.

10. The model cannot be reproduce, details are missing, data is also not available.

We are delighted to rectify this. Data and code were shared with the Reviewers in the Nature Research reporting summary (Software and Code Statement). In the statement, we pointed to <https://github.com/menicgiulia/MachineLearningFoodProcessing>.

In the meantime, we created a new GitHub repository with updated codes and data at <https://github.com/menicgiulia/MLFoodProcessing>

We have also updated the “Code and Data Availability Statement” in the manuscript to point directly to our repository. The Reviewer can find an overview of the code related to FoodProX and FPro in SI Section 8.

In summary, we would like to thank the Reviewer for inspiring us to expand our work on feature importance, making our analysis more thorough.

Reviewer #4 (Remarks to the Author):

Review, Mike Rayner, University of Oxford, 12th August 2022

General points

This paper introduces 'FoodProX, a machine learning classifier trained to predict the degree of processing of any food. Importantly, FoodProX allows {the authors} to define a continuous index that captures the degree of processing of any food, and can help quantify the overall diet quality of individuals, unveiling statistical correlations between the degree of processing characterizing individual diets and multiple disease phenotypes.' (Line 65) *It claims that 'The remarkable ability of the FoodProX predictions to replicate the manual NOVA classification confirms that food processing results in distinct patterns of nutrient alterations, accurately detected by machine learning.* (Line 315)

This is really interesting and useful paper that will make a major contribution to debates about the impact of food processing on health and on the usefulness of food profiling systems that are entirely or even partially based on the degree of processing of a food. The paper, in effect, proposes an alternative (FoodProX) to the NOVA classification system – the best known and most extensively studied food scoring systems based on the degree of processing of a food up until now.

FoodProX differs from NOVA in generating a continuous score for the degree of processing of a food whereas NOVA classifies foods into four groups: unprocessed food, culinary ingredients, processed foods, ultra-processed food. FoodProX is based on the nutrient content of a food whereas NOVA relies on a person (or persons) manually allocating a food to one of the four groups based on a large number of considerations. For example, as I understand it, a chocolate cake made in the home would not be considered an ultra-processed food NOVA whereas a chocolate cake made in a factory would. As the authors suggest this means that FoodProX has considerable advantages over NOVA.

We thank the Referee for the thorough and accurate summary of the manuscript. We are glad that the Reviewer found the manuscript interesting and useful, and remarked the advantages of our methodology compared to the state of the art. We modified the main text and supplementary material to address each of the valuable suggestions of the Reviewer.

1) My only reservation about this paper is there seems to be some circularity in the arguments. In particular since FoodProX was developed using a training dataset that had foods classified according to NOVA, the ability of the FoodProX predictions to replicate the manual NOVA classification is hardly 'remarkable' whereas Line 315 suggests that it is.

We are delighted to clarify this. First, it is important to stress that the creators of NOVA (Prof. Monteiro's group) do not use nutritional information to classify foods. Indeed, in the first lines of Monteiro et al. [14] we read: "NOVA is the food classification that categorizes foods according to the extent and purpose of food processing, rather than in terms of nutrients." NOVA researchers make their assessments by evaluating textual information, such as the list of ingredients, the food name, and description. Interestingly, despite the fact that NOVA researchers do not use nutrient concentrations to classify food, our algorithm leveraging nutrients as input is able to predict the known NOVA labels with extremely good precision (Lines 122-

123), suggesting that nutrients encode the bulk of information regarding NOVA classes, which is what we found to be remarkable. The accuracy by which we can predict the NOVA classes (e.g., AUC > 96% for all NOVA classes) is equally remarkable from the machine learning perspective: if the nutrients had not been informative, we would have found AUC~50% for all classes, i.e., the expected value for a random classifier.

2) I am also not convinced that the ability of FoodProX to replicate NOVA does 'confirm that food processing results in distinct patterns of nutrient alterations, accurately detected by machine learning. (Line 315 again). This conclusion suggests a causal relationship between processing (as defined by NOVA) and changes in nutrient levels. Of course this is likely to be the case, as demonstrated by the authors' persuasive example of the changes in nutrients between uncooked onion and processed onion. But this causal relationship is not, I think, demonstrated by the ability of FoodProX to replicate NOVA classifications. I think the authors mean something like: 'The classifications of the NOVA system are associated with distinct patterns of nutrients that can be detected by machine learning'.

The Reviewer is correct and we are sorry for the over-simplification. Following the Reviewer's suggestion, we have reformulated the sentence now at Lines 344-349. In particular, we have changed "food processing" to "NOVA classes".

3) However, on balance, I think this paper is really interesting in both its main findings and some of its 'incidental' finding such as 'Currently, our analysis in Section S1.5 shows that an unsupervised hierarchical clustering of foods, leveraging the widest nutrient panel available in FNDDS, can [sic] not able to independently reproduce the four NOVA classes'.

We are very pleased the Reviewer found our paper interesting and took the time to go through the supplementary materials.

4) I think the Discussion could be longer but I also rather like its brevity which leaves the reader the freedom to draw their own conclusions.

Thank you — We expanded and reformulated the Discussion following the Reviewer's suggestions.

Specific comments.

5) Line 46 'First, as the current categorization relies on expertise-based manual evaluation of each food, it covers only 35% of the foods catalogued by U.S. Department of Agriculture (USDA) (SI Section 1).' I recommend making this sentence clearer. What current categorisation are we talking about here? Presumably, the authors mean the NOVA classification system?

Correct — Thank you for pointing this out. We changed "the current categorization" with "NOVA"

6) Line 188 'In other words, while information on additives can improve FProX's performance, changes in the nutrient profile already carry the bulk of the predictive power.' This is an interesting observation but it begs the question of the extent to which information on changes in

the nutrient profile would improve an algorithm based on additives. And also whether it is the additives or the nutrient changes which make processed foods ‘unhealthy’. Perhaps this could be discussed in the Discussion.

This is a great suggestion! Prompted by the Reviewer’s question regarding a model purely based on additives, we trained and tested a new model that takes as input only the number of additives, and tries to predict NOVA classes. Compared to the initial model leveraging as input nutrients + number of additives, the new model drastically underperforms in terms of precision, significantly increasing the number of false positives for NOVA 1, 2, and 3 classes. Indeed, the area under the Precision-Recall curve (AUP) decreases 70.01% for NOVA 1, 97.05% for NOVA 2, and 50.96% for NOVA 3. For NOVA 4, AUP decreases 4.77%, suggesting that the presence of additives is a good predictor of ultra-processed food as defined by NOVA, but it misses the processing nuances represented in the other NOVA classes.

We added Lines 212-215 to the manuscript and provided a detailed description of the experiment in SI Section 6.

7) Line 315 ‘The remarkable ability of the FoodProX predictions to replicate the manual NOVA classification confirms that food processing results in distinct patterns of nutrient alterations, accurately detected by machine learning.’ I recommend deleting the word ‘remarkable’ and changing second half of sentence as discussed above.

We agree — Therefore, we changed accordingly.

8) Line S452 ‘Currently, our analysis in Section SI.5 shows that an unsupervised hierarchical clustering of foods, leveraging the widest nutrient panel available in FNDDS, can [sic] not able to independently reproduce the four NOVA classes.’ Change ‘can’ to ‘is’.

Thank you for catching this typo. We changed “can” to “is”, as suggested by the Reviewer.

In summary, we wish to thank for the many constructive and expert comments offered by the Reviewer. We are impressed by the depth of attention and understanding the Reviewer has offered.

Bibliography

- [1] B. Ravandi, P. Mehler, A.-L. Barabási, and G. Menichetti, “GroceryDB: Prevalence of Processed Food in Grocery Stores,” *medRxiv*, p. 2022.04.23.22274217, Apr. 2022.
- [2] X. Chen *et al.*, “Consumption of ultra-processed foods and health outcomes: A systematic review of epidemiological studies,” *Nutrition Journal*, vol. 19, no. 1. BioMed Central Ltd, pp. 1–10, 20-Aug-2020.
- [3] V. Braesco *et al.*, “Ultra-processed foods: how functional is the NOVA system?,” *Eur. J. Clin. Nutr.*, vol. 76, no. 9, pp. 1245–1253, Mar. 2022.
- [4] M. Beslay *et al.*, “Ultra-processed food intake in association with BMI change and risk of overweight and obesity: A prospective analysis of the French NutriNet-Santé cohort,” *PLoS Med.*, vol. 17, no. 8, 2020.
- [5] T. Fiolet *et al.*, “Consumption of ultra-processed foods and cancer risk: Results from NutriNet-Santé prospective cohort,” *BMJ*, vol. 360, 2018.
- [6] M. Adjibade *et al.*, “Prospective association between ultra-processed food consumption and incident depressive symptoms in the French NutriNet-Santé cohort,” *BMC Med.*, vol. 17, no. 1, pp. 1–13, 2019.
- [7] B. Srour *et al.*, “Ultra-processed food intake and risk of cardiovascular disease: Prospective cohort study (NutriNet-Santé),” *BMJ*, vol. 365, 2019.
- [8] B. Srour *et al.*, “Ultraprocessed Food Consumption and Risk of Type 2 Diabetes among Participants of the NutriNet-Santé Prospective Cohort,” *JAMA Intern. Med.*, vol. 180, no. 2, pp. 283–291, Feb. 2020.
- [9] J. Gehring *et al.*, “Consumption of Ultra-Processed Foods by Pesco-Vegetarians, Vegetarians, and Vegans: Associations with Duration and Age at Diet Initiation,” *J. Nutr.*, vol. 151, no. 1, pp. 120–131, Jan. 2021.
- [10] J. Konieczna *et al.*, “Contribution of ultra-processed foods in visceral fat deposition and other adiposity indicators: Prospective analysis nested in the PREDIMED-Plus trial,” *Clin. Nutr.*, vol. 40, no. 6, pp. 4290–4300, Jun. 2021.
- [11] S. L. Canhada *et al.*, “Ultra-processed foods, incident overweight and obesity, and longitudinal changes in weight and waist circumference: The Brazilian Longitudinal Study of Adult Health (ELSA-Brasil),” *Public Health Nutr.*, vol. 23, no. 6, pp. 1076–1086, Apr. 2020.
- [12] L. Wang *et al.*, “Association of ultra-processed food consumption with colorectal cancer risk among men and women: results from three prospective US cohort studies,” *BMJ*, vol. 378, p. e068921, Aug. 2022.
- [13] G. Menichetti and A.-L. Barabási, “Nutrient concentrations in food display universal behaviour,” *Nat. Food*, vol. 3, no. 5, pp. 375–382, May 2022.
- [14] C. A. Monteiro *et al.*, “NOVA The Food System,” *World Nutr.*, vol. 7, no. 7, pp. 1–3, 2016.
- [15] E. Martínez Steele, B. M. Popkin, B. Swinburn, and C. A. Monteiro, “The share of ultra-processed foods and the overall nutritional quality of diets in the US: Evidence from a nationally representative cross-sectional study,” *Popul. Health Metr.*, vol. 15, no. 1, Feb. 2017.
- [16] A. S. Baldrige *et al.*, “The healthfulness of the US packaged food and beverage supply: A cross-sectional study,” *Nutrients*, vol. 11, no. 8, 2019.
- [17] M. S. Da Silva Oliveira and L. Silva-Amparo, “Food-based dietary guidelines: A comparative analysis between the Dietary Guidelines for the Brazilian Population 2006

- and 2014,” *Public Health Nutr.*, vol. 21, no. 1, pp. 210–217, Jan. 2018.
- [18] R. Shepherd, “Resistance to Changes in Diet,” *Proc. Nutr. Soc.*, vol. 61, no. 2, pp. 267–272, May 2002.
- [19] M. P. Kelly and M. Barker, “Why is changing health-related behaviour so difficult?,” *Public Health*, vol. 136, pp. 109–116, Jul. 2016.
- [20] “Ultra-processed food and drink products in Latin America: Trends, impact on obesity, policy implications,” *Pan American Health Organization World Health Organization*, 17-Jun-2019. [Online]. Available: <https://iris.paho.org/handle/10665.2/51094>. [Accessed: 28-Sep-2022].
- [21] C. A. Monteiro, G. Cannon, J. C. Moubarac, R. B. Levy, M. L. C. Louzada, and P. C. Jaime, “The UN Decade of Nutrition, the NOVA food classification and the trouble with ultra-processing,” *Public Health Nutr.*, vol. 21, no. 1, pp. 5–17, 2018.
- [22] A. S. Baldrige *et al.*, “The Healthfulness of the US Packaged Food and Beverage Supply: A Cross-Sectional Study,” *Nutrients*, vol. 11, no. 8, p. 1704, 2019.
- [23] C. A. Monteiro, G. Cannon, M. Lawrence, M. L. Louzada Costa, and P. Pereira Machado, “Ultra-processed foods, diet quality, and health using the NOVA classification system,” 2019.
- [24] Haut Conseil de la Santé Publique, “Avis relatif aux objectifs de santé publique quantifiés pour la politique nutritionnelle de santé publique (PNNS) 2018-2022,” 2018.
- [25] T. Parr, K. Turgutlu, C. Csiszar, and J. Howard, “Permutation Feature Importance.” [Online]. Available: github.com/parrt/random-forest-importances.

REVIEWER COMMENTS

Reviewer #1 (Remarks to the Author):

No further comments.

Reviewer #2 (Remarks to the Author):

I wish to thank the authors for addressing my comments in their review. The objective of the study is clearer to me ('to define a continuous index (FPro) that captures the degree of processing of any food'). However, I don't feel the introduction justifies this objective enough, the hypothesis need clarification, and some new lines in the introduction need revision because it shows some misunderstanding of the NOVA classification. I understand it is not the main goal of the paper, but it would need a discussion on the methods to assess food processing in the literature, to better justify the need for a new tool as authors propose. Currently, the paper does not provide a solid and clear justification of the tool, and the criticisms of NOVA are not sound, based on the detailed elements below:

NOVA is categorical, not ordinal, and so NOVA is not the 'highest' degree of processing. NOVA is based on the degree AND purpose of food processing, not just the degree, and the definition is qualitative in nature (descriptive is not I think the right term). Perhaps a continuous measure would be easier to apply for policy, than a categorical one like NOVA?

I don't fully understand the hypothesis when thinking about the food matrix effect, because indeed raw and minimally processed foods are more nutrient-dense, but how do we consider the difference between naturally occurring and added nutrients to food as they are processed? Like fiber. What are the assumptions based on the proposed FPro measure about this? If author think that the degree of processing is enough than perhaps, they could explain why?

According to NOVA, foods containing at least a cosmetic additive, or a substance of non-culinary use is defined as ultra-processed food. The definition is hard to apply if the list of ingredients is not available indeed. When the list of ingredients is not available, then researchers need to make assumptions based on descriptors to classify foods using NOVA (for example, all 'commercial' cookies are classified as ultra-processed (this is a limitation). However, in lines 63-65, you write 'Given these data limitations, current approaches have classified all foods with at least one ultra-processed ingredient as ultra-processed'. This statement is incorrect, because this practice follows the definition, it is not a limitation due to lack of data.

The following sentence needs nuance: (lines 70-71): While NOVA allows for a more refined analysis, all foods within this class are considered to have identical health consequences'. Perhaps, mention that so far epidemiological and clinical studies have only focused on ultra-processed food as a group. NOVA claims that all ultra-processed foods share similar characteristics (but health impact is not part of these

characteristics) and their impact on health is assessed through research.

Reviewer #3 (Remarks to the Author):

Thank you for addressing all of my questions.

The paper is publishable in this form, having all details and information about the study.

Reviewer #4 (Remarks to the Author):

This paper is, still, and in my view an excellent paper, improved by taking on board the reviewer comments. The authors' Response to Referees Letter is comprehensive, measured and helpful.

I have no further suggestions for improvements to this paper. Although I am not entirely sure that the authors have addressed my concern that there seems to be some circularity in their argument that the ability of FoodProX to replicate the manual NOVA classification is remarkable given that FoodProX was developed using a training dataset that had foods classified according to NOVA. I don't think the way the Monteiro team make their manual assessment is relevant here is it? And if it is the team were sure to be using some knowledge of the nutritional composition of the foods.

I am very pleased that the authors have followed up my suggestion for discussing the extent to which information on nutrient composition would improve an algorithm based on additives by investigating whether, and if so how, an algorithm based purely on additive compares with FoodProX (Lines 212-215 and SI Section 6.)

Mike Rayner, University of Oxford

Reviewer #1 (Remarks to the Author):

No further comments.

We wish to thank the Referee for the many constructive observations and for suggesting an interpretability analysis of our epidemiological model, which has undoubtedly improved the quality of our manuscript.

Reviewer #2 (Remarks to the Author):

I wish to thank the authors for addressing my comments in their review. The objective of the study is clearer to me ('to define a continuous index (FPro) that captures the degree of processing of any food').

We thank the Reviewer for acknowledging our efforts in addressing the Reviewer's comments and we are delighted to read that the objective of the study is now clearer. In the following, we address the remaining recommendations of the Reviewer.

1) However, I don't feel the introduction justifies this objective enough, the hypothesis need clarification, and some new lines in the introduction need revision because it shows some misunderstanding of the NOVA classification. I understand it is not the main goal of the paper, but it would need a discussion on the methods to assess food processing in the literature, to better justify the need for a new tool as authors propose. Currently, the paper does not provide a solid and clear justification of the tool, and the criticisms of NOVA are not sound, based on the detailed elements below:

We thank the Reviewer for prompting us to provide a clearer justification for FPro. Following the Reviewer's suggestion, we now clarify the need for an algorithm like FPro in the introduction at Lines 35-45 and address the recommendations of the Reviewer below.

2) NOVA is categorical, not ordinal, and so NOVA is not the 'highest' degree of processing. NOVA is based on the degree AND purpose of food processing, not just the degree, and the definition is qualitative in nature (descriptive is not I think the right term). Perhaps a continuous measure would be easier to apply for policy, than a categorical one like NOVA?

We fully agree with the Referee that NOVA, as a classification system, is categorical rather than ordinal, especially when the purpose of food processing is taken into account. However, it is fair to say that NOVA *does aim to achieve a degree-based classification as well*. Indeed, the "extent of food processing" mentioned by Monteiro et al. in [1] implies an "extent of change from nature, ranging from unaltered foods in their original form to industrial products", as explained in [2]–[4]. Nevertheless, we fully agree with the Reviewer that a continuous score has major advantages for effective policy-making, compared to a categorical one. Indeed, the minimal food substitution strategy described in Lines 323-353 would have not been possible with NOVA. Hence, prompted by the Referee's suggestion, we changed "descriptive" to "qualitative" in Line 61.

3) I don't fully understand the hypothesis when thinking about the food matrix effect, because indeed raw and minimally processed foods are more nutrient-dense, but how do we consider the difference between naturally occurring and added nutrients to food as they are processed? Like fiber. What are the assumptions based on the proposed FPro measure about this? If author think that the degree of processing is enough than perhaps, they could explain why?

We are delighted to clarify this. Our algorithm does not assess one nutrient at a time, but learns from the patterns of correlated nutrient changes for a fixed amount of mass (100 grams). This implies that a single

high or low nutrient value does not uniquely determine the final FPro of a food, as the score will depend on the likelihood of observing the overall pattern of nutrient concentrations in unprocessed foods or in ultra-processed foods. For example, while fortified foods exhibit similar nutrient content to unprocessed foods for minerals and vitamins, the assessment of their overall nutrient profile by the algorithm identifies patterns of concentrations that are unlikely to be found in minimally processed whole foods, resulting in a high FPro.

To develop a better intuition on how FPro leverages the whole nutrient profile, and in particular Fiber content, as raised by the Referee, in Figure S10 Panel A we report the nutrient profile of POST cereals compared to ‘Wheat Bran Unprocessed’, and in Panel B we show the ratio of the nutrient values in cereals with their counterpart in wheat bran. Interestingly, the pattern of alterations involves all nutrients, increasing the level of FPro even for simple products like Post Shredded Wheat ‘N Bran, as its nutrient profile is not characteristic of any natural ingredient, but it corresponds to a mildly processed food (see Section S2.5). All foods in Figure S10 have a similar amount of Carbohydrates per 100g, but they differ in terms of Fiber, with ‘Wheat bran, unprocessed’ (FPro=0.0682) showing the highest amount. However, even if “Post Grape-Nuts” (FPro=0.9603) and “Post Shredded Wheat” (FPro=0.5685) follow with a comparable amount of Fiber, they exhibit a drastically different FPro.

4) According to NOVA, foods containing at least a cosmetic additive, or a substance of non-culinary use is defined as ultra-processed food. The definition is hard to apply if the list of ingredients is not available indeed. When the list of ingredients is not available, then researchers need to make assumptions based on descriptors to classify foods using NOVA (for example, all ‘commercial’ cookies are classified as ultra-processed (this is a limitation). However, in lines 63-65, you write ‘Given these data limitations, current approaches have classified all foods with at least one ultra-processed ingredient as ultra-processed’. This statement is incorrect, because this practice follows the definition, it is not a limitation due to lack of data.

We apologize for the oversimplification used in the previous formulation of our manuscript. Our intention was to reflect that the practice suggested by Monteiro et al. [2], [5], is a consequence of the lack of well-regulated data on food labels indicating food processes and their purpose, an essential piece of information to correctly implement NOVA. Following the Reviewer’s suggestion, we reformulated Lines 73-76.

5)The following sentence needs nuance: (lines 70-71): While NOVA allows for a more refined analysis, all foods within this class are considered to have identical health consequences’. Perhaps, mention that so far epidemiological and clinical studies have only focused on ultra-processed food as a group. NOVA claims that all ultra-processed foods share similar characteristics (but health impact is not part of these characteristics) and their impact on health is assessed through research.

We agree with the Reviewer and we changed the sentence accordingly.

In summary, we wish to thank the Referee for prompting us to clarify key aspects of our work, offering more nuance and hence more accuracy.

Reviewer #3 (Remarks to the Author):

Thank you for addressing all of my questions.

The paper is publishable in this form, having all details and information about the study.

We wish to thank the Reviewer for the useful clarifying comments on our modeling efforts, which have significantly improved the paper, making our pipeline more sound and robust.

Reviewer #4 (Remarks to the Author):

This paper is, still, and in my view an excellent paper, improved by taking on board the reviewer comments. The authors' Response to Referees Letter is comprehensive, measured and helpful.

We thank the Reviewer for finding the manuscript further improved by the Reviewers' comments, and for appreciating our "Response to Referees" Letter.

1) I have no further suggestions for improvements to this paper. Although I am not entirely sure that the authors have addressed my concern that there seems to be some circularity in their argument that the ability of FoodProX to replicate the manual NOVA classification is remarkable given that FoodProX was developed using a training dataset that had foods classified according to NOVA. I don't think the way the Monteiro team make their manual assessment is relevant here is it? And if it is the team were sure to be using some knowledge of the nutritional composition of the foods.

In Machine Learning (ML) the algorithm is always trained and tested on previously available data. At first, this approach indeed appears to be circular, so to avoid that, ML literature has thoroughly addressed it by developing multiple cross-validation techniques to avoid over-fitting and misleading high performances [6]. We utilized all these techniques to avoid the pitfalls of circular thinking, as noted by the Reviewer (see Lines 132-137 in the Results and SI Section 2.1). Yet, when dealing with real-world scenarios, most ML algorithms are lucky to achieve an AUC of 0.6-0.7, despite an extensive learning process. The low AUC is rooted in the fact that what they try to predict is simply too noisy to be reliably predicted [7]. What we call *remarkable* in the case of food processing is that we obtain an AUC > 0.96 for all NOVA classes, i.e., the algorithm can almost perfectly reproduce the human classification. Such a high performance is rarely seen in ML, allowing us to confidently apply the algorithm to other foods, that have not been manually classified before.

2) I am very pleased that the authors have followed up my suggestion for discussing the extent to which information on nutrient composition would improve an algorithm based on additives by investigating whether, and if so how, an algorithm based purely on additive compares with FoodProX (Lines 212-215 and SI Section 6.)

Mike Rayner, University of Oxford

This was indeed a great suggestion! Thank you for prompting us to clarify this aspect of our work, which has undoubtedly strengthened the interpretability of the algorithm.

In summary, we thank the Reviewer again for the depth of attention devoted to our manuscript, which has unquestionably benefited from it.

Bibliography

- [1] C. A. Monteiro *et al.*, "NOVA The Food System," *World Nutr.*, vol. 7, no. 7, pp. 1–3, 2016.
- [2] C. R. Sadler, T. Grassby, K. Hart, M. Raats, M. Sokolović, and L. Timotijevic, "Processed food classification: Conceptualisation and challenges," *Trends Food Sci. Technol.*, vol. 112, pp. 149–162, Jun. 2021.
- [3] A. Fardet, "Minimally processed foods are more satiating and less hyperglycemic than ultra-processed foods: a preliminary study with 98 ready-to-eat foods," *Food Funct.*, vol. 7, no. 5, pp. 2338–2346, May 2016.
- [4] C. A. Monteiro, "Nutrition and health. The issue is not food, nor nutrients, so much as processing," *Public Health Nutr.*, vol. 12, no. 5, pp. 729–731, May 2009.
- [5] C. A. Monteiro *et al.*, "Ultra-processed foods: What they are and how to identify them," *Public Health Nutr.*, vol. 22, no. 5, pp. 936–941, 2019.

- [6] P. Dangeti, *Statistics for machine learning : build supervised, unsupervised, and reinforcement learning models using both Python and R*, 1st editio. Birmingham, England ; Packt Publishing.
- [7] E. Guney, J. Menche, M. Vidal, and A.-L. Barábasi, “Network-based in silico drug efficacy screening,” *Nat. Commun.*, vol. 7, no. May 2015, p. 10331, 2016.

REVIEWERS' COMMENTS

Reviewer #2 (Remarks to the Author):

I thank the authors for addressing my comments, there are still some points for which I disagree with authors, such as the idea that NOVA classifies food according to the extent of processing, but I feel we have discuss this paper enough

Reviewer #2 (Remarks to the Author):

I thank the authors for addressing my comments, there are still some points for which I disagree with authors, such as the idea that NOVA classifies food according to the extent of processing, but I feel we have discuss this paper enough.

We thank the Reviewer for acknowledging our efforts in addressing their comments and for prompting us to clarify key aspects of our work. As discussed in the previous rounds of responses, in the first lines of Monteiro et al. [1] we read: “NOVA is the food classification that categorizes foods according to the extent and purpose of food processing, rather than in terms of nutrients.” According to other literature reviews and papers on the topic [2]–[4], the “extent of food processing” mentioned by Monteiro et al. in [1] implies an “extent of change from nature, ranging from unaltered foods in their original form to industrial products”. Furthermore, to avoid any ambiguities, FPro is calculated as a function of the probabilities for NOVA 1 and NOVA 4, the two extreme classes which are clearly ranked according to an increasing extent of processing. We now mention this aspect at Lines 164-165. We also explicitly state “extent and purpose of food processing” from [1] at Lines 48-49.

- [1] C. A. Monteiro *et al.*, “NOVA The Food System,” *World Nutr.*, vol. 7, no. 7, pp. 1–3, 2016.
- [2] C. R. Sadler, T. Grassby, K. Hart, M. Raats, M. Sokolović, and L. Timotijevic, “Processed food classification: Conceptualisation and challenges,” *Trends Food Sci. Technol.*, vol. 112, pp. 149–162, Jun. 2021.
- [3] A. Fardet, “Minimally processed foods are more satiating and less hyperglycemic than ultra-processed foods: a preliminary study with 98 ready-to-eat foods,” *Food Funct.*, vol. 7, no. 5, pp. 2338–2346, May 2016.
- [4] C. A. Monteiro, “Nutrition and health. The issue is not food, nor nutrients, so much as processing,” *Public Health Nutr.*, vol. 12, no. 5, pp. 729–731, May 2009.